# Adult microglial TGFβ1 is required for microglia homeostasis via an autocrine mechanism to maintain cognitive function in mice

Alicia Bedolla[1,2], Elliot Wegman[1], Max Weed [1], Messiyah K. Stevens [3], Kierra Ware [1], Aditi Paranjpe[4], Anastasia Alkhimovitch[1,5], Igal Ifergan[1,2,5], Aleksandr Taranov [1,2], Joshua D. Peter [1], Rosa Maria Salazar Gonzalez[6,7], J. Elliott Robinson [6,7], Lucas McClain[1], Krishna M. Roskin [5,7,8], Nigel H. Greig[9] & Yu Luo [1,2,5] ✉

While TGF-β signaling is essential for microglial function, the cellular source of TGF-β1 ligand and its spatial regulation remains unclear in the adult CNS. Our data supports that microglia but not astrocytes or neurons are the primary producers of TGF-β1 ligands needed for microglial homeostasis. Microglia-*Tgfb1* KO leads to the activation of microglia featuring a dyshomeostatic transcriptome that resembles disease-associated, injury-associated, and aged microglia, suggesting microglial self-produced TGF-β1 ligands are important in the adult CNS. Astrocytes in MG-Tgfb1 inducible (i)KO mice show a transcriptome profile that is closely aligned with an LPS-associated astrocyte profile. Additionally, using sparse mosaic single-cell microglia KO of TGF-β1 ligand we established an autocrine mechanism for signaling. Here we show that MG-*Tgfb1* iKO mice present cognitive deficits, supporting that precise spatial regulation of TGF-β1 ligand derived from microglia is required for the maintenance of brain homeostasis and normal cognitive function in the adult brain.

Microglia are commonly known as the resident immune cells in the central nervous system (CNS), but their roles expand beyond that of innate immunity. At homeostasis, microglia play a variety of regulatory roles such as surveilling the brain parenchyma for injury or disease, phagocytosis, and synaptic pruning[1–4]. In addition to their homeostatic role, microglia are vital in inflammatory response initiation and regulation. In the case of injury or inflammation, microglia dynamically alter their function on a spectrum of activation states ranging from the more pro-inflammatory M1-like state to the anti-inflammatory M2-like state[5–7]. Previous studies have shown that transforming growth factor beta (TGF-β) signaling is required for the development of microglia during the embryonic stage[8]. Specifically, a cleverly designed "CNS-

[1]Department of Molecular and Cellular Biosciences, University of Cincinnati, Cincinnati, OH, USA. [2]Neuroscience Graduate Program, University of Cincinnati, Cincinnati, OH, USA. [3]Department of Psychology, Vanderbilt University, Nashville, TN, USA. [4]Information Services for Research, Cincinnati Children's Hospital Medical Center, Cincinnati, USA. [5]Division of Immunobiology, Cincinnati Children's Hospital Medical Center, University of Cincinnati College of Medicine, Cincinnati, OH, USA. [6]Division of Experimental Hematology and Cancer Biology, Department of Pediatrics, Cincinnati Children's Hospital Medical Center, Cincinnati, OH, USA. [7]Department of Pediatrics, University of Cincinnati College of Medicine, Cincinnati, US. [8]Division of Biomedical Informatics, Cincinnati Children's Hospital Medical Center, Cincinnati, USA. [9]Translational Gerontology Branch, Intramural Research Program, National Institute on Aging, National Institutes of Health, Baltimore, MD, USA. ✉e-mail: luoy2@ucmail.uc.edu

specific" *Tgfb1* knockout (KO) mouse model was developed by over-expressing the *Tgfb1* gene in T-cells (via an Il-2 promoter) in a global *Tgfb1* KO mouse model, which depletes CNS TGF-β1 constitutively but partially compensates peripheral TGF-β1 levels[8]. Using this mouse model, it was reported that in the absence of TGF-β1 in the CNS during development, microglia do not establish their signature gene expression, indicating that TGF-β1 is required for normal microglial development[8]. While this study supported the importance of TGF-β1 in microglial development, whether TGF-β signaling is required in mature microglia to maintain their survival and function in the adult brain is not known. Moreover, serum levels of TGF-β1 in this "CNS" *Tgfb1* KO mouse model were undetectable[8], resulting in a potential confound due to altered TGF-β1 levels in peripheral tissues and serum that could have indirect effects on microglia maturation.

Regarding whether TGF-β signaling is required for the maintenance of homeostasis in adult microglia, there is some controversy in the literature. Buttgereit et al. [9] reported that *Sall1*^CreER*Tgfbr2*^fl/fl mice showed activated morphology in microglia, accompanied by upregulation of CD45/CD11c and certain inflammatory cytokines (Il1b, TNF, and Cxcl10). However, Arnold et al. reported no phenotype in microglia in Cx3cr1^CreER*Tgfbr2*^fl/fl mice treated with tamoxifen (TAM) at an age of 30 days[10]. Recently, another study reported an intermediate phenotype (termed "primed") on evaluating the *Cx3cr1*^CreER-*Tgfbr2*^fl/fl mice[11]. Whereas they established that TGF-β signaling, via TGF-βR2 in adult microglia, is necessary for maintaining the ramified morphology and certain features of microglial homeostasis[11], they further reported that inducible knockout (iKO) of *Tgfbr2* in adult microglia only leads to a "primed" state in microglia without effects on many microglia homeostatic signature genes such as *P2ry12*, *Tmem119*, *Hexb*, and *Sall1*[11]. These studies, while supporting the importance of TGF-β signaling in microglial maturation during developmental stages and maintaining certain features of homeostasis in adulthood, also generate standing questions regarding the requirements and the degree of importance of TGF-β signaling in maintaining microglia homeostatic signature gene expression in the adult CNS. More importantly, the precise cellular source of TGF-β ligands, the spatial/temporal regulation of components of the TGF-β signaling pathway across different cell types in the CNS, and their functional relevance have yet to be identified, leaving some major gaps in our knowledge regarding the regulation of this important signaling pathway in the adult brain.

While it is assumed that many different cell types can be sources of TGF-β ligands in the CNS at homeostasis[12–17], the actual production of TGF-β ligands in different cell types in the brain has not been established. Additionally, two previous studies hint at a highly precise spatially localized regulation of the activation of TGF-β ligand through the interaction of leucine-rich repeat protein (LRRC33) and αVβ8 integrin[13,18,19]. However, whether the TGF-β ligand is regulated by a diffusible paracrine mechanism locally, or whether the ligand is more strictly regulated via an autocrine manner in the CNS is not known. The present study aims to address these major gaps in the field in this study.

Herein, using cell-type-specific conditional or iKO models of the *Tgfb1* gene, we demonstrate that microglia- but not neuron- or astrocyte-derived TGF-β1 ligand is required for the maintenance of homeostatic microglia in the early postnatal and adult brain. Furthermore, the loss of microglia-derived TGF-β1 ligands leads to the presence of reactive astrocytes within the brain and causes cognitive deficits in adult mice. Additionally, our study shows that the total TGF-β1 ligand level in the brain is substantially lower compared to that of serum or peripheral tissue, and that the adult brain has accordingly established a precise spatially controlled mechanism to regulate ligand production to maintain homeostasis in individual microglia that is dependent on microglial autocrine TGF-β signaling. Our data also show that TGF-β1 is enriched in microglia, whereas TGF-β2 is instead enriched in astrocytes. Following *Tgfb1* gene deletion in microglia, *Tgfb3* is

upregulated in the *Tgfb1* KO microglia. However, at least up to 12 weeks following global microglial *Tgfb1* deletion, neither the astrocytic *Tgfb2* nor the upregulated *Tgfb3* in microglia is able to compensate for the loss of function of *Tgfb1* or rescue the dyshomeostatic phenotype in *Tgfb1* KO microglia, suggesting distinct expression and functions of the different ligands in the CNS. We also address the questions of microglia-astrocyte crosstalk and the functional relevance of microglial TGF-β signaling in the adult CNS. With the importance of TGF-β signaling becoming more recognized in injury, neurodegeneration, and aging in the CNS, our study provides insights into the mechanisms of how TGF-β signaling can be regulated on a single-cell level via a microglia autocrine mechanism in the adult CNS and expands directions for future studies in understanding how TGF-β1 ligand production and downstream signaling in recipient cells can occur under these conditions.

## Results
### Abolishing TGF-β1 ligand expression specifically in CNS macrophages
To identify the cell type(s) in the CNS that provide TGF-β1 ligand to microglia and other TGF-β1 responsive cells, we first examined scRNAseq data sets published in previous studies[20–24]. Highly enriched *Tgfb1* mRNA levels in adult mouse microglia are observed in multiple scRNAseq datasets[20,21]. Further analysis of these data shows that astrocytes, neurons, and oligodendrocytes have minimal *Tgfb1* expression. Whereas oligodendrocyte precursor cells (OPCs) and endothelial cells have detectable levels of *Tgfb1* expression, these are substantially lower compared to microglia *Tgfb1* mRNA levels (Supplementary Fig. 1A, B)[20,21]. By searching published human scRNAseq datasets, we found that several independent human studies showed similar enrichment of *Tgfb1* gene expression in microglia (Supplemental Fig. 1C, D)[22–24] while one other independent study showed *Tgfb1* enriched expression in microglia only in AD patients but not in control microglia[25]. To validate microglia *Tgfb1* expression in the mouse brain, we undertook a combined RNAscope/IHC analysis to examine the cellular expression pattern of *Tgfb1* in the adult mouse brain. Our data show that, indeed, *Tgfb1* mRNA is enriched in IBA1-positive microglia but is not detected in neurons (Supplementary Fig. 1E). *Tgfb1* mRNA is also detected in a small population of non-IBA1+ cells, which could be endothelial cells or other glial cells. mRNA for the type 1 TGF-β1 receptor (TGF-βR1 or ALK5) is also detected in microglia, supporting that microglia can be both ligand-producing and responding cells for TGF-β1 signaling. To examine whether microglia are a major contributor to TGF-β1 production in the brain, we depleted microglia from the adult mouse brain using the well-established receptor for colony-stimulating factor 1 (CSFR1) antagonist PLX5622 (1200ppm in diet) following our previously published protocol (Supplementary Fig. 1F), which resulted in more than 90% of microglia ablation reported by us and other studies[26–28]. After successful microglia ablation, we examined the total *Tgfb1* mRNA in cortical tissue from the control or microglia-ablated mouse brain. qRT-PCR analysis shows that the PLX5622 treatment leads to a substantial depletion of microglia in the adult brain, indicated by a significant decrease of *Iba1* mRNA levels in brain tissue (<5% of WT levels), accompanied by a 70% decrease in total *Tgfb1* mRNA levels (Supplementary Fig. 1G). Since microglia compose only 5–10% of total brain cells[29,30], microglia ablation leading to a 70% decrease in total *Tgfb1* mRNA levels in the brain supports that microglia are a major component for TGF-β1 ligand production.

Next, to establish the direct functional relevance of microglia-produced TGF-β1 ligand to microglia homeostasis, the *Cx3cr1*^CreER line[31] was crossed with the *Tgfb1*^fl/fl line to enable TAM-induced TGF-β1 ligand loss in microglia in adulthood. To confirm the efficiency of *Tgfb1* gene deletion in microglia in the inducible MG-*Tgfb1* KO mice, in our recent study, we sorted microglia using an R26-YFP (yellow fluorescent protein) reporter allele (which labels ~90% of total microglia in the adult

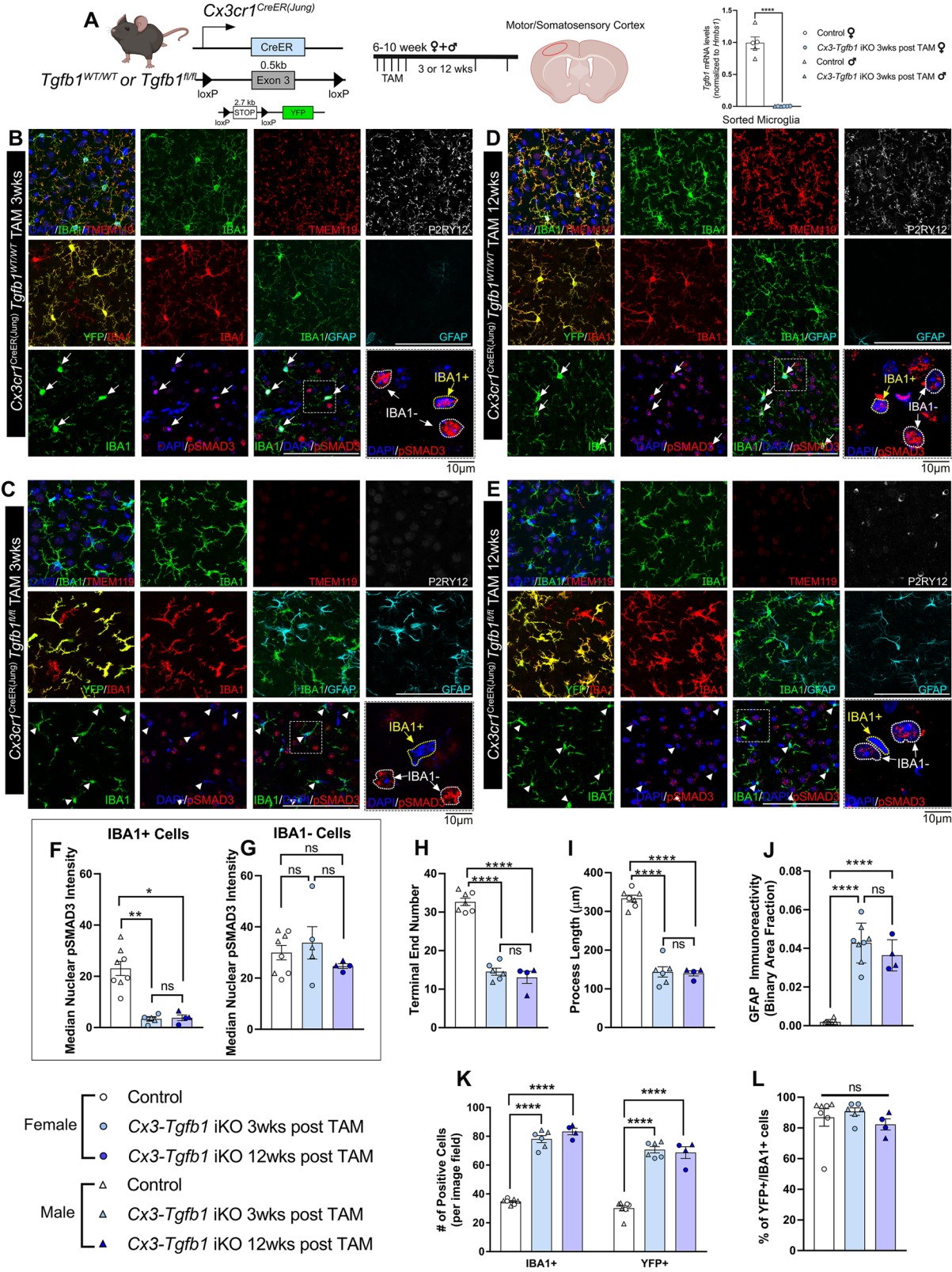

mouse brain) at 3 weeks following TAM treatment in $Cx3cr1^{CreER}Tgfb1^{wt/wt}R26\text{-}YFP$ and $Cx3cr1^{CreER}Tgfb1^{fl/fl}R26\text{-}YFP$ mice[32]. We demonstrate a significant decrease (99.8%) in $Tgfb1$ mRNA levels from sorted YFP+ microglia in $Cx3cr1^{CreER}Tgfb1^{fl/fl}$ mice in comparison to control microglia ($Cx3cr1^{CreER}Tgfb1^{wt/wt}$)[32] (Fig. 1A, right). To examine whether peripheral serum or tissue TGF-β1 levels are also affected in our MG-$Tgfb1$ iKO mice, we measured the TGF-β1 protein levels in serum and spleen using

enzyme-linked immunosorbent assay (ELISA) analysis. In contrast to the previous "CNS-$Tgfb1$" constitutive KO mouse model in which serum TGF-β1 levels were completely abolished[8], our $Cx3cr1^{CreER}Tgfb1$ iKO mice show no difference in serum or spleen TGF-β1 protein levels between Cx3cr1[CreER] WT and $Cx3cr1^{CreER}Tgfb1^{fl/fl}$ mice at 3 weeks after TAM treatment (Supplemental Fig. 2), confirming minimal interference in peripheral TGF-β1 ligand levels. Our ELISA results also show

**Fig. 1 | Microglia-specific *Tgfb1* gene deletion in a Cx3cr1<sup>CreER(Jung)</sup> driver line results in a loss of homeostasis of microglia and an increase in reactive astrocytes in the cortex of the adult mouse brain. A** (left) Mouse model for targeting microglial *Tgfb1* and experimental timeline and (right) indicates the gene deletion efficiency in FACS-isolated microglia. **B–E** Representative images for immunohistochemistry stained for IBA1, TMEM119, P2RY12, YFP, GFAP, and pSMAD3 in **B** control *Cx3cr1<sup>CreER(Jung)</sup>Tgfb1*<sup>wt/wt</sup> + TAM animals, **C** *Cx3cr1<sup>CreER(Jung)</sup>Tgfb1*<sup>fl/fl</sup> mice + TAM at 3 weeks after TAM administration, **D** control *Cx3cr1<sup>CreER(Jung)</sup>Tgfb1*<sup>wt/wt</sup> + TAM animals, and **E** *Cx3cr1<sup>CreER(Jung)</sup>Tgfb1*<sup>fl/fl</sup> mice + TAM at 12 weeks after TAM administration. **F, G** Quantification for pSMAD3 fluorescent intensity from **F** IBA1+ and **G** IBA1− cells. **H, I** Microglial morphological quantification of the **H** terminal end number, and **I** the summation of process lengths. **J** GFAP immunoreactivity quantified by binary area fraction. **K** total IBA1+ or YFP+ cells and **L** percentage of YFP+ cells among total

IBA1+ cells showing no change in % of YFP+ cell even 12 weeks after induction of *Tgfb1* KO. **A** Right ($n = 5$ for control and $n = 6$ for KO, $p = 0.0004$, Welch's *t*-test, two-sided); **F, G** ($n = 8$ for control, $n = 5$ for KO 3 week and $n = 4$ for KO 12 week, $**p = 0.006$ and $*p = 0.029$, Kruskal–Wallis test, Dunn's multiple comparisons test); **H, I, K, L** ($n = 7$ for control, $n = 6$ for KO 3 week and $n = 4$ for KO 12 week) and **J** ($n = 6$ for control, $n = 8$ for KO 3 week and $n = 4$ for KO 12 week). Each data point represents the average of a single animal, and the sex of each animal is indicated in the figure legend. Mean ± SE, Scale bar = 100 μm unless otherwise noted. **H–L** One-way ANOVA, Tukey's multiple comparisons test. $*p < 0.05$, $**p < 0.01$, $****p < 0.0001$. **A** Left, was created with BioRender.com released under a Creative Commons Attribution-NonCommercial-NoDerivs 4.0 International license. Source data are provided as a source data file.

that, compared to the high levels of TGF-β1 ligand in the spleen and serum (control and iKO spleen = 5.7 pg/ug or 5.9 pg/ug protein: control and iKO serum = 1.9 pg/ug or 1.8 pg/ug protein), total brain TGF-β1 levels are dramatically lower (below the detect limit of the ELISA assay), a result that is consistent with a recent independent study[33]. This suggests that TGF-β ligand production and release in blood/peripheral organs might be distinctly different from that of CNS given the substantial difference in their abundance. Due to the limitation of this analysis method, we were not able to confirm the loss of TGF-β1 ligand at a protein level in our *Cx3cr1<sup>CreER</sup>-Tgfb1* iKO mice directly using an ELISA assay in the light of the low total brain TGF-β protein levels. We next attempted to use fluorescence-activated cell sorting (FACS) analysis on surface TGF-β1 levels which demonstrated similar results. FACS analysis also showed substantially higher cell surface TGF-β1 expression in splenic myeloid cells, as compared to brain myeloid cells (Supplementary Fig. 2). This finding further supports that CNS TGF-β1 levels are substantially lower compared to peripheral tissue, making detection of the TGF-β1 protein in the CNS a challenge using either ELISA or FACS analysis. Our FACS analysis showed the detection of TGF-β1 protein on the surface of CD11b+/CD45+ splenocytes but no difference between control and *Cx3cr1<sup>CreER</sup>Tgfb1* iKO mice in this population. This corroborates that our iKO mouse model does not affect TGF-β1 expression in peripheral myeloid cells. However, since CD11b+/CD45+ cells from the brain show very low levels of TGF-β1 antibody binding we could not confidently compare the level of surface TGF-β1 expression on brain myeloid cells between the control and *Cx3cr1<sup>CreER</sup>-Tgfb1*<sup>fl/fl</sup> iKO mice. Nevertheless, qRT-PCR analysis from sorted brain myeloid cells demonstrates that the *Cx3cr1<sup>CreER</sup>Tgfb1*<sup>fl/fl</sup> transgenic line can efficiently delete the *Tgfb1* gene in these cells[32] with minimal interference on systemic serum or spleen TGF-β1 levels. Additionally, RNAseq of the sorted WT and *Tgfb1* iKO CNS myeloid cells confirms the loss of the floxed exon 3 from *Tgfb1* mRNA in iKO mice while not affecting mRNA counts of exon 4, which is downstream of the 3' loxP site (Supplementary Fig. 2). qRT-PCR analysis using an exon 3-specific primer/probe set also validated the efficient *Tgfb1* gene recombination in sorted microglia in iKO mice (Fig. 1A, right).

To further circumvent the challenge of direct detection of low levels of TGF-β1 protein in the CNS, we analyzed the downstream effector of TGF-β signaling, i.e., nuclear-localized phosphorylated SMAD3 (pSMAD3) protein levels. Co-immunohistochemistry analysis shows that in control mice (*Cx3cr1<sup>CreER+/−</sup>Tgfb1*<sup>wt/wt</sup> + TAM), pSMAD3 is detected in both IBA1+ microglia cells and IBA1− cells in the brain (Fig. 1A, right and B, E, bottom row). MG-specific deletion of the *Tgfb1* gene leads to an explicit and significant decrease of pSMAD3 immunoreactivity exclusively in (Ionized calcium-binding adaptor molecule 1) IBA1+ microglia, without affecting the pSMAD3 immunostaining in IBA1− cells (bottom rows of Fig. 1B–E and quantification in F, G). This specific loss of TGF-β1 signaling (pSMAD3) in microglia from MG-*Tgfb1* iKO mice confirms that microglia TGF-β1 signaling depends on microglia produced TGF-β1 ligand that cannot be compensated for by other cells.

## Microglial TGF-β1 is required for maintaining adult glia homeostasis

Next, we evaluated whether the loss of microglia-derived TGF-β1 ligand affects microglia morphology and homeostatic status in adult mice (6–10-week-old females and males). At 3 and 12 weeks post TAM administration, substantial morphological changes in microglia in MG-*Tgfb1* iKO mice were observed, as compared to control mice (Fig. 1B–E, IBA1 staining). This change in IBA1+ cell morphology in the iKO mice suggests loss of ramification and potential activation of microglia, which prompted us to perform additional detailed morphological analysis and an examination of the homeostatic microglia signature genes. Following microglial *Tgfb1* KO, microglia in *Cx3cr1<sup>CreER(Jung)</sup> Tgfb1* iKO mice showed less ramification, as indicated by decreased process terminal end number (Fig. 1H, control = 33, 3 week iKO = 15, 12 week iKO = 13) and total branch length (Fig. 1I, control = 330 μm, 3 week iKO = 149 μm, 12 week iKO = 140 μm) versus control microglia. Moreover, *Tgfb1* iKO microglia demonstrated a reduced expression of homeostatic microglia signature genes such as *P2ry12* and *Tmem119* (Fig. 1B–E, P2RY12, and TMEM119 staining). These results align with an impaired microglial homeostatic status, and are consistent with another independent study using *Sall1<sup>CreER</sup>Tgfbr2*<sup>fl/fl</sup> mice that showed similar morphological changes but did not examine the expression of *P2ry12* and *Tmem119*[9]. Notably, our observed phenotype is more severe than that of two previous studies in which adult microglial TGF-β signaling was abolished via KO of TGF-βR2[10,11]. This could be due to the short distance between the two loxP sites in such *Tgfb1* floxed mice, making it highly efficient for the recombination of the *Tgfb1* gene[32,34,35]. These results support the notion that microglial TGF-β1 signaling relies on TGF-β1 ligand produced by microglia, and a loss of microglia-derived TGF-β1 ligand cannot be compensated by TGF-β1 ligand production in other cell types in the adult CNS. Additionally, YFP reporter tracking in the *Cx3cr1<sup>CreER(Jung)</sup> Tgfb1*<sup>fl/fl</sup>*-R26YFP* and *Cx3cr1<sup>CreER(Jung)</sup> Tgfb1*<sup>wt/wt</sup>*-R26YFP mice* show that the total number of IBA1+ or YFP+ cells in the brain both increased in the iKO mice (Fig. 1K), while the % of YFP+ among IBA1+ cells in control versus iKO mice at 3- or 12-weeks post TAM remain the same (Fig. 1L). This suggests that the majority of the recombined microglia remain in the CNS and are likely not replaced by YFP- infiltrating monocytes during the weeks following TAM treatment. An increase in reactive astrocytes (indicated by upregulated Glial fibrillary acidic protein (GFAP) expression, Fig. 1B–E, quantification in Fig. 1J) was also observed both at 3- and 12-weeks after TAM treatment in the *Cx3cr1<sup>CreER(Jung)</sup>Tgfb1*<sup>fl/fl</sup> mice, as compared to the control *Cx3Cr1<sup>CreER(Jung)</sup>Tgfb1*<sup>wt/wt</sup> mice. Two independent *Cx3cr1<sup>CreER</sup>* mouse drivers were used in this study to confirm the phenotypes. We observed similar phenotypes in the *Cx3cr1<sup>CreER(Litt)</sup>Tgfb1*<sup>fl/fl</sup> mice at 5 and 8 weeks after TAM treatment, but not in the *Cx3Cr1<sup>CreER(Litt)</sup>Tgfb1*<sup>wt/wt</sup> control mice (Supplemental Fig. 3). A similar phenotype for both microglia and astrocytes is observed globally in other brain regions as well (Supplementary Fig. 4 showing hippocampus as another example, note that unlike cortical astrocytes that are mostly GFAP-, many hippocampal astrocytes are already GFAP+ at homeostasis in WT mice).

Abolishing TGF-β1 ligand expression in neonatal mice (TAM treatment at P3-P5 in $Cx3cr1^{CreER(Jung)}Tgfb1^{fl/fl}$ iKO) generates a similarly activated microglial phenotype with a reduced ramification morphology as well as significant loss of homeostatic microglia markers throughout the brain (Supplementary Fig. 5, showing representative images in the somatosensory cortical region). This suggests that microglia-derived TGF-β1 ligand is required to maintain homeostasis in the early post-natal mouse brain as well. We observed similar phenotypes in both male and female iKO mice both in neonatal and adult stages (Fig. 1 and Supplementary Fig. 5, no apparent sex differences are noticed, and the sex of each animal is indicated in the figures). Our results are consistent with a recent independent study investigating the origin of TGF-β1 ligands during embryonic development[36].

Both microglia and astrocytes are activated in the adult MG-$Tgfb1$ iKO mice. It is hence unclear whether microglia-derived TGF-β1 ligand loss leads to activation of both cell types directly, due to diminished signaling in each cell type, or whether such loss affected one cell type that indirectly activated the other cell type. To answer this question, we generated microglia-specific or astrocyte-specific TGF-β type 1 receptor ($Alk5$) iKO mice. Our data show that, loss of TGF-β signaling in microglia (via deletion of $Alk5$ gene) leads to an activation of microglia, which present with a loss of ramified morphology as well as decreased expression of homeostatic signature genes ($P2ry12$ and $Tmem119$) and an increased expression of activated microglia marker ($CD68$) (Fig. 2A–G). When we deleted TGF-β receptors in microglia, astrocytes were also activated in the MG-$Alk5$ iKO mice (Fig. 2C, G), which suggests that crosstalk between microglia and astrocytes in the $Tgfb$ ko MG is responsible for the activation of astrocytes. Indeed, when $Alk5$ is deleted in astrocytes (Fig. 2H, right) via the $mGfapCre$ driver[37], neither astrocytes show an increase in GFAP expression phenotype nor microglia show a loss of ramified morphology or decreased homeostatic marker expression in this astrocytic-Alk5 cKO mouse line (Fig. 2H–M).

## Astrocytic TGF-β1 is not necessary for glial homeostatic maintenance

Previous studies suggest that TGF-β ligands produced by astrocytes may play important roles in blood-brain barrier formation, stabilization, and maturation, as well as neuroprotection following injury or disease[38–40]. We, therefore, next investigated whether astrocyte-specific deletion of the $Tgfb1$ gene would likewise lead to alterations in microglia morphology and gene expression changes, such as $P2ry12$, $Tmem119$, and $Cd68$. Additionally, we investigated whether loss of the astrocytic $Tgfb1$ gene could also induce changes in astrocyte reactivity as observed in $Cx3cr1^{CreER}Tgfb1$ or $Cx3cr1^{CreER}Alk5$ iKO mice. To target adult astrocytes, the well-characterized $Aldh1l1^{CreER}$ mouse strain[41] was crossed with $Tgfb1^{fl/fl}$ mice to generate astrocytic $Tgfb1$ iKO mice ($Aldh1l1^{CreER}Tgfb1^{fl/fl}$). At 8 weeks following TAM administration, microglia morphology and astrocyte state were analyzed. Microglial morphology remained unchanged in $Aldh1l1^{CreER}Tgfb1^{fl/fl}$ animals, as compared to control mice ($Aldh1l1^{CreER}Tgfb1^{wt/wt}$ mice or $Tgfb1^{fl/fl}$ mice) (Fig. 3A–I for cortex and Supplementary Fig. 6 for hippocampus as an example). Additionally, no changes were observed in homeostatic microglia signature genes ($P2ry12$ or $Tmem119$, Fig. 3H, I). Nor did we observe upregulation of CD68 in microglia or an increase in GFAP expression in astrocytes when comparing the $Aldh1l1^{CreER}Tgfb1^{fl/fl}$ animals to WT controls at 8 weeks after TAM treatment (Fig. 3 F, G and Supplementary Fig. 6). This result further confirms that astrocytic TGF-β1 production is not required for the maintenance of microglia homeostatic morphology, P2RY12/TMEM119 expression, and suppression of astrocyte GFAP expression.

Additionally, we generated constitutive astrocytic $Tgfb1$ KO mice using the $mGfap^{cre}$ driver line[37], which targets a large population of astrocytes constitutively starting from neonatal stages[37] to, thereby, ensure that a greater population of astrocytes (95% of cortical astrocytes labeled with Ai14 reporter)[42] will have the TGF-β1 ligand KO. Comparing the morphology of $mGfap^{Cre}Tgfb1^{fl/fl}$ microglia to WT controls, no changes were evident in the ramification of microglia in this independent astrocytic-$Tgfb1$ cKO mouse line (Supplementary Fig. 7 for cortex and Supplementary Fig. 8 for hippocampus). We next examined the expression of homeostatic microglia signature genes $P2ry12$ and $Tmem119$, and did not observe any difference between $mGfap^{cre}Tgfb1^{fl/fl}$ mice and control mice (Supplementary Figs. 7 and 8). CD68 expression in microglia and astrocytic GFAP expression also were unchanged. These results, together, support the notion that under normal physiological conditions, adult astrocytes do not produce TGF-β1 ligand necessary for homeostatic maintenance in microglia or quiescence in astrocytes regarding GFAP upregulation. It remains to be determined whether other cellular or functional changes in the astrocytes are altered. Similarly, it is not clear whether, under injury or pathological conditions, astrocytes could upregulate TGF-β1 ligand to modulate glial responses to injury or neurodegeneration.

## Forebrain excitatory neurons do not produce TGF-β1 for microglia

Next, we investigated whether neurons are an additional source for $TGF$-β1 ligand production for adult microglia. To this end, we generated a forebrain excitatory neuron-specific $Tgfb1$ iKO mouse model. To target forebrain neurons, a $Camk2a^{CreER}$ line[43] was crossed with the $Tgfb1^{fl/fl}$ line to induce TGF-β1 ligand KO in excitatory neurons[43]. $Camk2a^{CreER}$ has been reported to recombine in a widespread manner in the cortex, hippocampus, and striatum[43]. Eight weeks after TAM administration, microglial morphology remained unchanged in the $Camk2a^{CreER}Tgfb1^{fl/fl}$ mice compared to WT controls (Fig. 3J–R for cortex and Supplementary Fig. 9 for hippocampus). Additionally, no alterations in TMEM119 or P2RY12 expression in microglia nor an increase in GFAP expression was observed. This supports the notion that microglia do not rely on neuronal TGF-β1 ligands to maintain homeostasis in the adult brain.

## Mosaic MG-specific Tgfb1 KO results in patchy activation of glia

The $Cx3cr1^{CreER}$ line has previously been reported to also target border-associated macrophages (BAMs) that reside in the pia and vasculature[32,44]. Additionally, the two $Cx3cr1^{CreER}$ mouse lines replace the endogenous $Cx3cr1$ gene with the cre expression cassette, resulting in heterozygosity for the $Cx3cr1$ gene in both controls ($Cx3cr1^{CreER}Tgfb1^{wt/wt}$) and in iKO ($Cx3cr1^{CreER}Tgfb1^{fl/fl}$) mice[31,45]. In an effort to further improve the targeting specificity for parenchymal microglia, the $Tmem119^{CreER}$ and $P2ry12^{CreER}$ mouse lines have recently been generated, utilizing homeostatic microglia signature gene promoters to drive CreER cassette expression without affecting the endogenous gene expression of $Tmem119$ or $P2ry12$[46,47]. Our and other recent studies report that the $Tmem119^{CreER}$ and the $P2ry12^{CreER}$ mouse lines show less TAM-independent "leaky" recombination events, but have the drawback of a lower recombination efficiency, which leads to only a subset of microglia being recombined in these two mouse lines (based on R26-YFP reporter gene expression and qRT-PCR results in sorted microglia, Fig. 4A, right)[32,44]. Two copies of the CreER alleles in the homozygous in the $P2ry12^{CreER(mut/mut)}$ mouse lines[47] increased the total recombination efficiency but not to the extent of the $Cx3cr1^{CreER}$ line (Fig. 4A, right)[32,45]. To examine whether mosaic gene deletion of the TGF-β1 ligand in a subset of parenchyma microglia would generate any phenotype in microglia, we crossed $Tmem119^{CreER}$ and $P2ry12^{CreER}$ (heterozygous cre or homozygous cre) with the $Tgfb1^{fl/fl}$ line to generate different parenchymal-microglia specific $Tgfb1$ iKO lines with variety degree of mosaic MG-$Tgfb1$ deletion.

To investigate whether partial deletion of microglial $Tgfb1$ could still produce the disruption of microglial homeostasis we observed in the $Cx3Cr1^{CreER}Tgfb1$ iKO lines, we next evaluated microglial morphology in $Tmem119^{CreER}$ (heterozygous cre) and $P2ry12^{CreER}$ (heterozygous

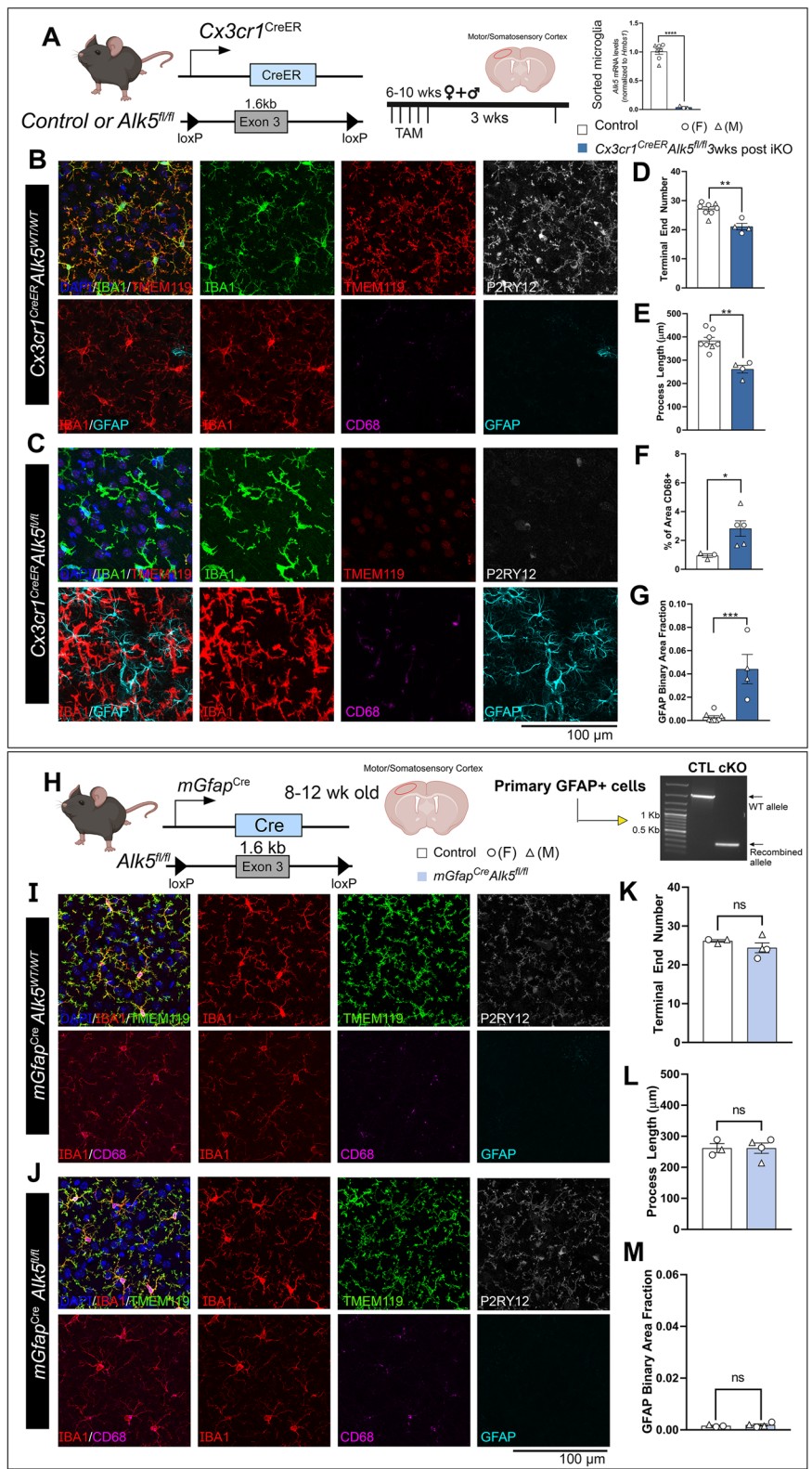

and homozygous cre) *Tgfb1* iKO mice. Consistent with the mosaic recombination of the R26-YFP reporter gene and the partial reduction of *Tgfb1* mRNA levels in adult microglia in the *Tmem119*^CreER and *P2ry12*^CreER mouse line (Fig. 4A, right), 5 weeks after TAM administration, we observed distinct islands of microglia patches with an activated morphology in both the *Tmem119*^CreER *Tgfb1* and *P2ry12*^CreER *Tgfb1* iKO mice (Fig. 4B–D) with a varying degree of % of activation in microglia consistent with the degree of *Tgfb1* gene deletion

efficiency (Fig. 4A, right, E)[32]. In addition to the mosaic and patchy morphological changes, we also observed a decrease in morphological ramification and expression of homeostatic microglia signature genes *P2ry12* and *Tmem119* in these patches of microglia (Fig. 4B, C, quantification Fig. 4F–H). With two copies of the CreER alleles in the *P2ry12*^CreER(mut/mut)*Tgfb1*^fl/fl iKO mice, the phenotype is more severe with >50% of the microglia losing TMEM119 expression (Fig. 4E) and moderate but significant degree of TMEM119 protein downregulation

**Fig. 2 | ALK5-dependent TGF-β signaling is required for the maintenance of microglial homeostasis but not for astrocytic homeostasis. A** (left) Mouse model for targeting microglial TGF-β type 1 receptor *Alk5* and experimental timeline. (Right) indicates the gene deletion efficiency in FACS-isolated microglia (*n* = 7 control, *n* = 3 iKO, *p* < 0.0001). **B, C** Representative immunohistochemistry images of IBA1, TMEM119, P2RY12, CD68, and GFAP in the cortex of **B** control animals and **C** *Cx3cr1*^CreER(Jung)^*Alk5*^fl/fl^ knockouts 3 weeks after TAM administration. Quantification of **D** microglial process terminal end numbers, **E** total microglial process length, (**D, E** *n* = 8 control, *n* = 4 iKO, *p* = 0.001 for (**D**) and *p* = 0.0004 for (**E**)), **F** % of CD68 immunoreactive positive area (*n* = 3 control, *n* = 5 iKO, *p* = 0.0396), and **G** GFAP immunoreactive positive area fraction (*n* = 8 control, *n* = 4 iKO, *p* = 0.002, Mann Whitney test, two-sided). **H** (left) Mouse model for targeting astrocytic *Alk5* and experimental timeline. (right) Indicates the gene recombination efficiency in cultured GFAP+ primary cells isolated from control or cKO mice using a PCR primer set

that flanks the entire loxP-Alk5-loxP cassette. Cells from three different cKO mice show similar results. **I, J** Representative immunohistochemistry images of IBA1, TMEM119, P2RY12, CD68, and GFAP in the cortex of **I** control animals, **J** *mGfap*^Cre^*Tgfb1*^fl/fl^ constitutive knockouts at 12-weeks-old. Quantification of **K** microglial process terminal end numbers, **L** total microglial process length, and **M** GFAP immunoreactive positive area fraction (**K**–**M** *n* = 3 control, *n* = 4 cKO). *\*p* < 0.05, \*\**p* < 0.01, \*\*\**p* < 0.001, ns = not significant. Mean ± SE, All panels are analyzed by two-sided Student's *t*-test, except panel (**G**). (>40 Microglia were quantified for each animal and the average from one mouse was plotted as a single data point in the figure panel and treated as *n* = 1 for statistical analysis). Scale bar = 100 μm. **A, H**, Left, were created with BioRender.com and released under a creative commons attribution-noncommercial-noderivs 4.0 International license. Source data are provided as a source data file.

(Fig. 4F) and partial loss of ramification (Fig. 4G, H) even in the adjacent TMEM119+ patches. The morphological changes and loss of *P2ry12* and *Tmem119* expression were not observed in control *Tmem119*^CreER^*Tgfb1*^wt/wt^ or *P2ry12*^CreER^*Tgfb1*^wt/wt^ TAM mice (Supplementary Fig. 10). To investigate whether *Tgfb1* KO microglia die over time and whether the activated IBA1+ cells are instead infiltrated monocytes, we crossed the *R26YFP* reporter allele in these mouse lines. Our data show that total IBA1+ cells increased in the *P2ry12*^CreER(mut/mut)^*Tgfb1*^fl/fl^ iKO mice, consistent with a more severe phenotype in this line that affects a higher population of microglia (Supplementary Fig. 10B and 10D). However, the percentage of YFP+ cells among total IBA1+ cells remains the same in WT and iKO mice in all the above three mouse lines (Supplementary Fig. 10A–C, quantification panel E, and panel G). Notably, not all iKO microglia would be expected to be labeled by YFP reporter due to the independence of recombination in distinct floxed alleles, especially given that the floxed *Tgfb1* allele is much shorter than the R26-YFP cassette. However, the activated YFP+ microglia patches and the similar % of YFP+ microglia in these microglia-specific *CreER* lines support the notion that it is, indeed, the parenchymal microglia that become activated and are not replaced by infiltrating YFP-negative peripheral monocytes.

We also observed increased activation of astrocyte patches (visualized by GFAP immunoreactivity) in the cortex of these mosaic microglia-*Tgfb1* ligand iKO mouse brains (Fig. 4B–D). On a populational level (analyzed using the entire imaging field), we observed a moderate correlation between the degree of microglial activation and the astrocyte GFAP immunoreactivity (Fig. 4I, $R^2 = 0.2432$, $p = 0.0523$ and J, $R^2 = 0.2518$, $p = 0.0476$), consistent with our hypothesis that loss of TGF-β1 ligand from microglia leads to loss of homeostasis in microglia which, in turn, activates astrocytes. To further investigate whether there is a close spatial correlation between local dyshomeostatic microglia and activated astrocytes, we analyzed each individual patch of dyshomeostatic TMEM119- microglia (area of the individual patch and number of dyshomeostatic microglia in each patch) and the local astrocytes activation (GFAP immunoreactivity within the individual patches) or GFAP+ astrocytes in physical contact with the dyshomeostatic microglia territory (measured by the number of GFAP+ astrocytes that are in contact with the individual dyshomeostatic TMEM119− microglia patches). Our data show that there is a positive correlation with all of the parameters analyzed regarding the local astrocyte activation in relationship to local mosaic microglia activation (Fig. 4K $R^2 = 0.6815$, $p < 0.0001$, Fig. 4L $R^2 = 0.5769$, $p < 0.0001$, Fig. 4M $R^2 = 0.2482$, $p < 0.0001$), supporting precise local crosstalk between microglia and astrocytes.

**Sparse individual iKO microglia reveal an autocrine signaling mechanism**

Notably, in the *Tmem119*^CreER^*Tgfb1*^fl/fl^ and *P2ry12*^CreER^*Tgfb1*^fl/fl^ heterozygous CreER mice (which had mosaic *Tgfb1* gene deletion) the distinct mosaic patches of the few *Tgfb1* KO microglia that showed altered

morphology and loss of expression of homeostatic microglia signature genes are surrounded by wildtype microglia cells which can produce the TGF-β1 ligand (Fig. 4). This raises the interesting question of whether individual microglia rely on self-produced TGF-β1 ligand that is secreted in an autocrine manner or whether individual microglia could utilize the TGF-β1 ligand from neighboring microglia via a paracrine mechanism.

To further investigate the spatial resolution of TGF-β1 ligand production by individual microglia and investigate whether individual microglia rely on self-produced TGF-β1 ligand using an autocrine mechanism, we designed a mosaic sparse recombination strategy using the *Cx3cr1*^CreER(Jung)^*Tgfb1*^fl/fl^ line with a titrated TAM dilution. To accomplish sparse recombination, we first tested it in the *Cx3cr1*^CreER^R26-YFP reporter line. Utilizing this reporter mouse line, we tested the TAM dosage of 1:50 and 1:7–1:10 of the concentration (180 mg/kg) that is utilized in our full dose recombination. Our results (Supplementary Fig. 11) show that for both the 1:50 and 1:7–1:10 TAM dosages, we observed very sparse YFP+ cells in the parenchyma that are also P2RY12+ (suggesting sparse recombination can occur in parenchymal microglia instead of BAMs). This result supports the feasibility of inducing sparse gene deletion in individual microglia surrounded by WT microglia using a titration of TAM dosage. Since the 1:7–1:10 dosage range provided recombination events that were sufficiently sparse, we carried out our subsequent experiments with this range of dosage. A diluted dose of 1:7 (25 mg/kg) of TAM was given over the course of 3 days to *Cx3cr1*^CreER(Jung)^*Tgfb1*^fl/fl^R26-YFP or control mice. This resulted in sparse labeling of microglia in the *Cx3cr1*^CreER(Jung)^ line, allowing for single-cell analysis of whether microglia depend on self-secreted TGF-β1 ligands to maintain their homeostasis. Remarkably, at 2–3 weeks post TAM administration, we observed isolated sparse individual IBA1+ cells in the parenchyma of the *Cx3cr1*^CreER(Jung)^*Tgfb1*^fl/fl^R26-YFP TAM-treated mice (Fig. 5C, G <10% of total IBA1+ cells) that presented with altered morphology (less ramified, Fig. 5C, J, and K), accompanied by a decrease in TMEM119 expression (Fig. 5C, H blue bar and Supplementary Video 1). These sparse mutated single IBA1+ cells are in the brain parenchyma and do not show typical blood vessel-associated macrophage morphology (Fig. 5C). However, the percentage of YFP+/TMEM119−/IBA1+ cells is very low (Fig. 5F), suggesting that at this low level of TAM dosage, on a single cell level, the recombination of the R26-YFP reporter allele can happen independently from the deletion of targeted floxed genes, a result that is also supported by our recent study using dual reporter alleles in microglia[32]. In the *Cx3cr1*^CreER^*Tgfb1*^wt/wt^R26-YFP TAM-treated mice, we observed a similar frequency of sparse YFP+ microglia (Fig. 5E) but did not find any individual non-BAM microglia that show this phenotype of activated microglia (marked by loss of TMEM119 expression and altered morphology, Fig. 5B, G), suggesting that the sparse individual "activated" microglia in the *Cx3cr1*^CreER^*Tgfb1*^fl/fl^ mice is due to loss of TGF-β1 ligand in the single sparse microglia.

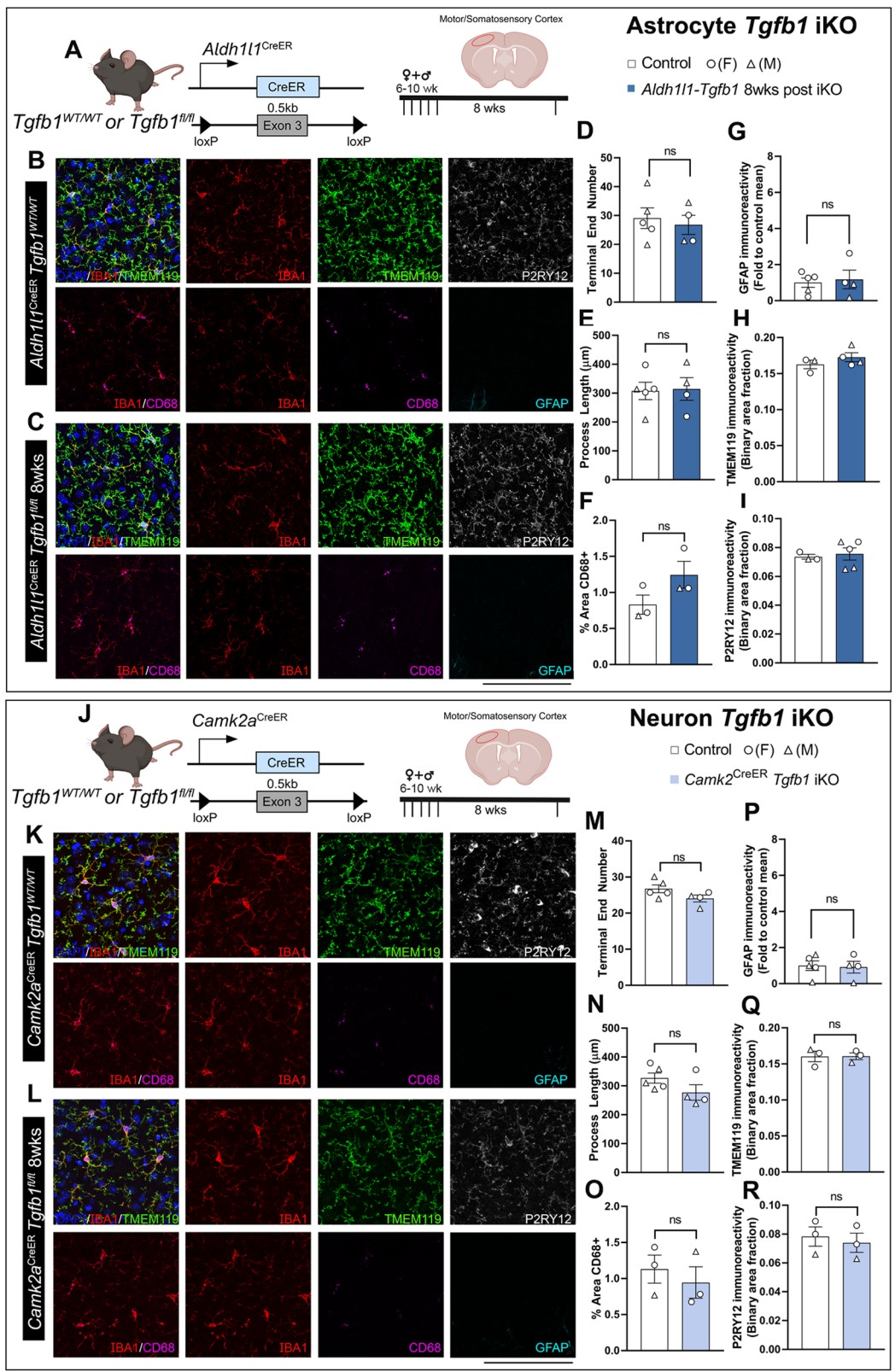

To further confirm this hypothesis, we carried out a combined immunohistochemistry (IHC)/RNAscope using the same *Tgfb1* RNAscope probe we used to confirm microglial TGF-β1 expression (Fig. 6A, B). Indeed, we observed significantly decreased *Tgfb1* RNAscope probe hybridization in the individual microglia that specifically showed altered morphology accompanied by the loss of TMEM119 expression, while surrounding IBA1+/TMEM119+ microglia showed normal TGF-β1 RNAscope signal (Fig. 6B, C). We further carried out immunostaining for the detection of the downstream signaling effector of TGF-β1 signaling (pSMAD3) in the mosaic sparse MG-*Tgfb1* iKO brain and confirmed that loss of pSMAD3 is detected specifically in sparse individual microglia that are TMEM119- and morphologically altered (Fig. 6D, E and Supplementary Fig. 12 for additional image examples). Therefore, our results suggest that microglia show a

**Fig. 3 | Astrocyte-specific or forebrain neuronal specific *Tgfb1* gene deletion in the *Aldh1l1*[CreER] or *Camk2a*[CreER] drivers does not affect the homeostasis of microglia or GFAP expression in astrocytes in adult mouse brain (cortex).**
**A** Astrocyte iKO mouse model and experimental timeline. **B, C** Representative immunohistochemistry images of cortex from TAM treated (8 weeks post) control **B** *Aldh1l1*[CreER]*Tgfb1*[wt/wt] and **C** iKO *Aldh1l1*[CreER]*Tgfb1*[fl/fl] tissue showing IBA1, TMEM119, P2RY12, CD68, and GFAP immunostaining. Quantification of microglia ramification via **D** process terminal end numbers, **E** total process length, and **F** % of CD68+ immunoreactive area. **G** Quantification of astrocyte reactivity using GFAP immunoreactive positive area fraction, quantification of **H** TMEM119, and **I** P2RY12 immunoreactivity. (**D, E, G**, *n* = 5 control, *n* = 4 iKO) and (**F, H, I**, *n* = 3 control and *n* = 3, 4, and 5 for iKO). **J** Neuronal iKO mouse model and experimental timeline. **K, L** Representative images of TAM treated (8 weeks post) control

*Camk2a*[CreER]*Tgfb1*[wt/wt] (**K**) and iKO *Camk2*[CreER] *Tgfb1*[fl/fl] (**L**) tissue showing IBA1, TMEM119, P2RY12, CD68, and GFAP immunoreactivity. Quantification of microglia ramification via **M** process terminal end number, **N** total process length, and **O** CD68+ immunoreactive % area. **P** Quantification of astrocyte reactivity using GFAP+ immunoreactive area fraction, and quantification of **Q** TMEM119 and **R** P2RY12 immunoreactivity (**M, N, P**, *n* = 5 control and *n* = 4 iKO) and (**O, Q, R**, *n* = 3 for both control and iKO). Mean ± SE (>40 microglia were quantified for each animal and the average from one mouse was plotted as a single data point in the figure panel and treated as *n* = 1 for statistical analysis). ns = not significant. Two-sided Student's *t*-test, scale bar = 100 μm. **A, J** Created with BioRender.com and released under a Creative Commons Attribution-NonCommercial-NoDerivs 4.0 International license. Source data are provided as a source data file.

precise spatial regulation of autocrine TGF-β1 signaling reliant on self-produced TGF-β1 ligands under homeostatic physiological conditions. We next asked whether this loss of homeostasis in individual *Tgfb1* KO microglia at 2–3 weeks post TAM is sustained with time or whether, with surrounding wildtype microglia, the individual *Tgfb1* KO microglia can regain homeostasis. At 8 weeks post TAM treatment in sparse MG-*Tgfb1* iKO mice, the phenotype of sparse TMEM119 negative and morphologically altered microglia is no longer observed (Fig. 5D, F, I). This suggests that these sparse *Tgfb1* KO microglia can potentially regain homeostasis (indicated by normal ramified morphology and restoration of TMEM119 expression) at 8 weeks after TAM treatment in a sparse mosaic MG-*Tgfb1* gene deletion model. Due to the inability of the R26-YFP reporter allele to simultaneously label the very sparse *Tgfb1* KO microglia, we are not able to use the YFP reporter to track the individual *Tgfb1* KO microglia longitudinally. However, in the full dosage of TAM-treated Cx3cr1CreER-Tgfb1 iKO mice, we do not see decreased YFP+ cells due to cell death up to 12 weeks after TAM. Additionally, we observed sustained activated phenotype in the majority of the microglia (YFP+) at up to 12 weeks post TAM treatment (Fig. 1E), suggesting that when the majority of the microglia lose TGF-β1 ligand it could be more difficult to recover compared to individual sparsely activated microglia.

### MG-Tgfb1 iKO leads to dyshomeostasis glial transcriptomics

To further characterize the transcriptional changes following the loss of microglial TGF-β1 ligand, microglia, and astrocytes were sorted from the *Cx3cr1*[CreER(Jung)]*Tgfb1*[fl/fl]*R26-YFP* and *Cx3cr1*[CreER (Jung)]*Tgfb1*[wt/wt]*R26-YFP* animals 3 weeks after TAM administration based on YFP expression or ASCA2 immunolabeling (astrocyte staining, Supplementary Figs. 13 and 7)[48] and subjected to RNAseq analysis. The purity of samples collected using this sorting method is validated by qRT-PCR for microglia and astrocytic signature genes, respectively (Supplementary Fig. 13). We sorted brain microglia based on recombined YFP reporter expression (which labels about 90% parenchyma microglia in the whole brain) instead of CD11b+/CD45[low] to avoid the potential caveat that loss of TGF-β signaling in microglia may increase CD45 expression[9] and selectively enrich for a subpopulation of microglia in the KO brain. Also, due to the immunostaining data showing the downregulation of homeostatic markers such as P2RY12 or TMEM119 in the KO microglia, we could not use these markers to distinguish microglia vs BAMs. Therefore, we gated CNS myeloid cells based on the YFP signal, which could contain both CNS microglia and BAMs. Principal component analysis (PCA) shows that wildtype microglia samples and *Tgfb1* iKO microglia distinctively clustered together (Supplementary Fig. 14). The heatmap shows significantly differentially expressed genes (fold change ≥ |1.5| and adj. *p* value < 0.05, Supplementary Fig. 14 and Supplementary Data 1 and 2). In contrast to a recent study using Cx3cr1[CreER]*Tgfbr2*[fl/fl] receptor inducible KO mice, which reported no changes in many homeostatic microglia signature genes in KO mice[11], we observed a large set of differentially expressed genes including downregulation of many microglia homeostatic

signature genes (*P2ry12*, *Tmem119*, *Sall1*, etc. Fig. 7C) and upregulation of immune response regulating genes (*TNF*, *Il1b*, interferon responsive genes). Using gene set enrichment analysis (GSEA) we observed the upregulation of several pathways related to immune response, immune cell recruitment, and interferon response (Supplemental Fig. 15 and Supplementary Data 3 and 4). We also observed downregulation in platelet aggregation pathway genes (Supplemental Fig. 15). For astrocytes, we observed upregulation of multiple LPS-associated astrocytic genes (*Serping1*, *Ifit3*, *Gbp3*, Fig. 7D, H) and we also observed an increased interferon response (*Irf7*, *Irf9*, Fig. 7D). Consistently, GSEA analysis also showed increased interferon activity and decreased metabolic functions (NADH, mitochondria, acetyl CoA, Supplemental Fig. 16 and Supplementary Data 5 and 6) which suggests a transition from metabolic support functions to an activated pro-inflammatory state in astrocytes from the MG-*Tgfb1* iKO brain. These data suggest that microglia and astrocytes had disrupted homeostatic functional activity after the loss of microglial TGF-β1.

Recently, in a human cellular context, Abud et al. have characterized the transcriptomic profile of human microglial-like cells (iMGLs) derived from human iPSCs that underwent 24 h of TGF-β ligand withdrawal[49], which represent human TGF-β signaling targets that are in the acute stage and are potentially direct targets of the signaling pathway. Given that several independent human scRNAseq datasets suggest similar enrichment of TGF-β signaling components in human microglia[22–24], we analyzed our dataset and the human TGF-β-deprived iMGLs dataset and identified overlapping genes significantly upregulated or downregulated in the absence of TGF-β signaling in both datasets. We identified 2390 upregulated and 2324 downregulated genes in Cx3cr1[CreER]*Tgfbr2*[fl/fl] mice, and there were 1962 upregulated and 1199 downregulated genes reported in the iMGL dataset[49]. From these genes, we identified 237 upregulated and 147 downregulated genes common to both our dataset and the iMGL dataset (Fig. 7E, F). Gene ontology (GO) analysis was performed for both the upregulated and downregulated gene lists. The upregulated genes were enriched for genes involved in cytokine signaling, inflammatory response, small GTPase and NF-kappa B signaling pathways, response to TNF, and regulation of T cells (Fig. 7E and Supplementary Data 7). The downregulated genes were enriched for genes involved in dendritic cell chemotaxis and migration, macrophage migration, cellular defense response, positive regulation of type II IFN production, and inflammatory response (Fig. 7F and Supplementary Data 7). A number of chemokine receptors are downregulated in the absence of TGF-β signaling (*Ccr1*, *Ccr2*, *Ccr5*, *Ccr6*, and *Cx3cr1*) while a number of chemokine ligands are upregulated (*Ccl5*, *Ccl24*, *Cxcl1*, *Cxcl9*, *Cxcl10*, and *Cxcl16*). The presence of chemokine-related genes in both lists explains why there is some overlap in GO terms between the upregulated and downregulated gene lists.

We next wanted to compare the transcriptomic profile of the *Cx3cr1*[CreER]*Tgfb1*[fl/fl] microglia and astrocytes in relation to previously characterized non-homeostatic microglial states. For microglia, we censured multiple previous studies to generate a list of signature genes

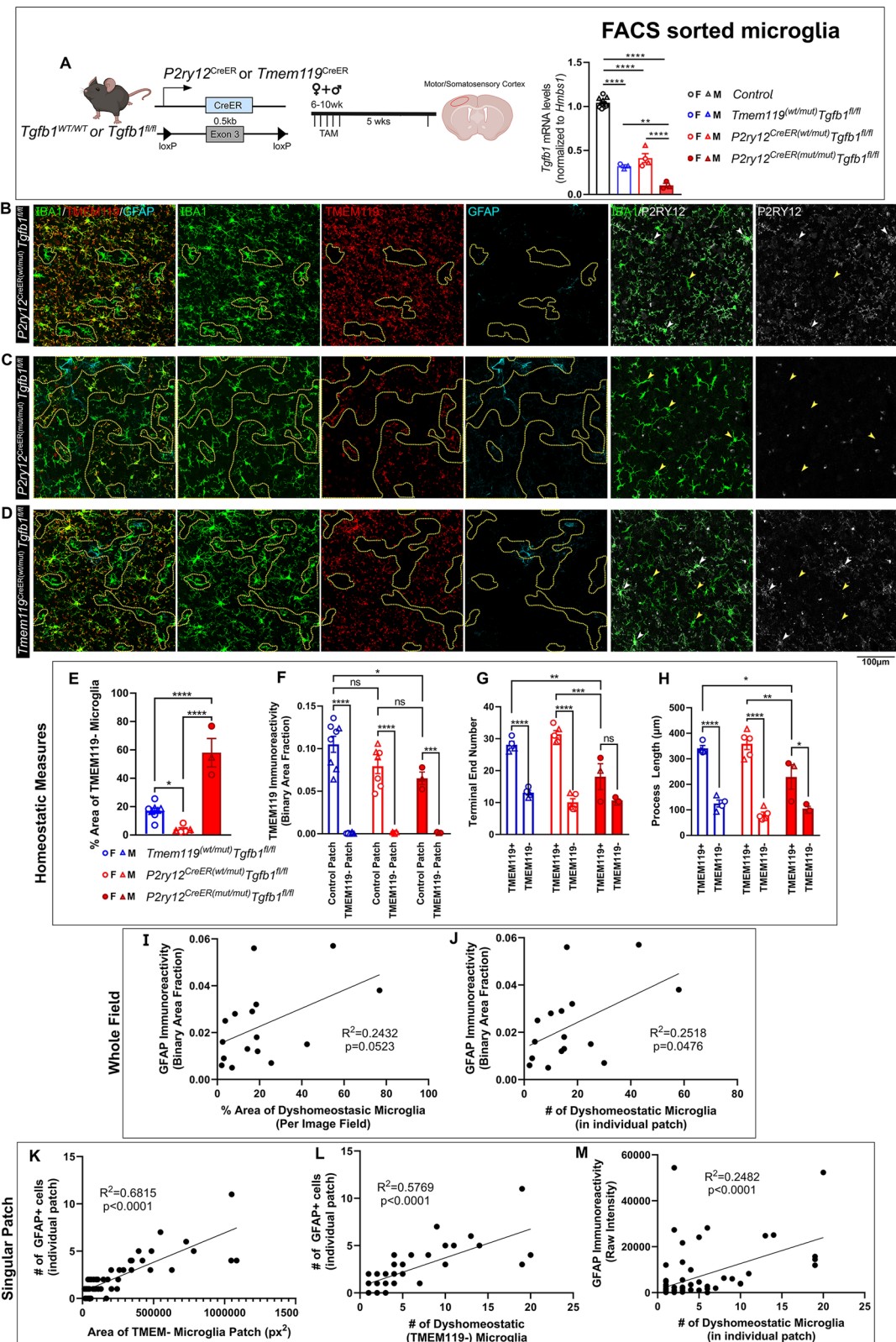

associated with aging, CNS injury (Traumatic brain injury-TBI), and amyloid-beta pathological conditions[50–54]. We observed that three weeks after the loss of microglial TGF-β1 ligand, microglia showed downregulation of microglial homeostatic genes (*Hexb*, *P2ry12*, *Tmem119*, *Cst3*, *Cd33*, *Cx3cr1*, Fig. 7G) suggesting dyshomeostasis. We also observed increased expression of aging microglia signature genes (ex. *Ifitm3*, *Ccl12*, *Il1b Ccl2*, *Lgals3*), and injury-associated TBI signature

genes such as *Irf7*, *Igf1*, *Cxcl10*, *Ccl12*, *Axl*, *Cd63*, and *Cybb* (Fig. 7G). Recently amyloid beta-induced microglial transcriptomic changes have been profiled into DAM 1 and DAM 2 stages, the transition of which depends on TREM2 signaling[50–53]. *Tgfb1* iKO microglia resembles the upregulation of a subset of amyloid beta-associated microglia profile genes, while showing downregulation of other amyloid-beta profile genes. Upon further examination, we noted that the

**Fig. 4 | Mosaic deletion of the *Tgfb1* gene in subsets of parenchymal microglia in the *P2ry12*^creER^*Tgfb1*^fl/fl^ or the *Tmem119*^CreER^*Tgfb1*^fl/fl^ iKO mice leads to distinct patches of dyshomeostatic microglia in the adult mouse brain. A** (left) *P2ry12*^CreER^ or *Tmem119*^CreER^ mouse driver to induce *Tgfb1* KO in P2RY12+ or TMEM119+ microglia and experimental timeline. **A** (right) Indicates the gene deletion efficiency in FACS-isolated microglia in all the different mouse lines (*n* = 10, 3, 4, 3 for each group presented, \*\*\*\*<0.0001 and \*\**p* = 0.0038). TAM treated, **B** *P2ry12*^CreER(wt/wt)^*Tgfb1*^fl/fl^, **C** *P2ry12*^CreER(mut/mut)^*Tgfb1*^fl/fl^, and **D** *Tmem119*^CreER(wt/mut)^*Tgfb1*^fl/fl^ iKO representative images showing immunohistochemistry for IBA1, TMEM119, P2RY12, and GFAP. Yellow dotted outlines indicate microglia regions with downregulated TMEM119 expression. White arrows depict homeostatic IBA1+ cells that still express P2RY12 expression. Yellow arrowheads show IBA1+ cells that are no longer expressing P2RY12. **E** Quantification of % area of TMEM− regions across all three lines in the whole image field (*n* = 8, 5, 3 for each group presented, \*\*\*\*<0.0001 and \**p* = 0.0312, panel **A**, right and **E** are analyzed by one way ANOVA, two-sided, Tukey's multiple comparisons). **F** Quantification of TMEM119 expression in *P2ry12*^CreER(wt/mut)^*Tgfb1*^fl/fl^,

*P2ry12*^CreER(mut/mut)^*Tgfb1*^fl/fl^, and *Tmem119*^CreER^*Tgfb1*^fl/fl^ (*n* = 8, 7, 3 for each group presented, \**p* = 0.0162, \*\*\**p* = 0.0008, and \*\*\*\*<0.0001). **G, H** Quantification of microglia morphology across the three different mouse lines for **G** terminal number and **H** process length (*n* = 4, 5, 3 for each group, \**p* = 0.0306 and 0.0237, \*\**p* = 0.0078 and 0.0016, \*\*\**p* = 0.0002 and \*\*\*\**p* < 0.0001. **F–H** Analyzed by two-way ANOVA, two-sided, Tukey's multiple comparison). **I, J** Correlation of total GFAP immunoreactivity vs % area or a number of dyshomeostatic microglia based on TMEM− the area from representative images across all 3 mouse lines. **K–M** Correlations of different parameters within an individual dyshomeostatic patch comparing **K** number of GFAP+ cells vs % area of TMEM− microglia, **L** number of GFAP+ cells vs number of TMEM− dyshomeostatic microglia, and **M** GFAP immunoreactivity vs a number of dyshomeostatic TMEM− microglia. Mean ± SE, for correlations, a simple linear regression was used for analysis. Scale bar = 100 μm. **A** Left, was created with BioRender.com released under a Creative Commons Attribution-NonCommercial-NoDerivs 4.0 International license. Source data are provided as a source data file.

---

upregulated genes in iKO microglia represent DAM 1 signature genes (*B2m*, *Apoe*, *Tyrobp*, but a decrease in *Trem2 levels*) while downregulated genes in iKO microglia represent DAM 2 signature genes (*Ccl6*, *Cst7*, *Cd9*, *Csf1*, *Itgax*) which correlates well with the downregulation of TREM2 in iKO microglia (Fig. 7G). Additionally, after the loss of microglial TGF-β1, consistent with the observed reactivity in astrocytes by upregulation of GFAP protein, we observed downregulation of some astrocytic homeostasis genes (*Aldh1l1*, *Acsl6*, *Aldoc*), upregulation of in vivo LPS-associated genes (*Gbp3*, *Gbp2*, *Gbp6*, *Psmb8*, Fig. 7H), and no discernable changes in the ischemia-associated genes in astrocytes from MG-*Tgfb1* iKO brains (Fig. 7H).

To further analyze potential ligand-receptor signaling that could mediate the microglia-astrocytes crosstalk and interactions, we next analyzed potential ligands from microglia, potential receptors from astrocytes, and potential target genes in astrocytes-target genes that are differentially expressed in our RNAseq dataset. NicheNet package v1.1.1 in R v4.0.2 was implemented to infer ligand-receptor interactions using 1754 and 100 differentially expressed genes (FC ≥ |1.5| and FDR < 0.5) from microglia and astrocytes, respectively. The curated ligand-receptor interactions from the NicheNet database were used as a reference. We identified seven main ligand targets from microglia (Jam2, Apoe, Trf, Vcam1, Lgals3, Adam17, Adam9) that could be interacting with two astrocyte receptors (Itgb1 and Ldlr) (Supplementary Fig. 17). The downstream gene targets that showed the greatest difference after TGF-β1 ligand loss were Dusp10, Lmna, Cd320, Ldlr, Mfge8, Nek8 and Tm4sf1 in astrocytes. These ligand, receptor, and gene target changes are potential TGF-β dependent pathways that are used in the activation of astrocytes by microglia.

Transcriptomic data from microglia and astrocytes also reveal interesting expression patterns of signaling components of the TGF-β signaling pathways. Consistent with qRT-PCR data from sorted microglia and astrocytes (Supplementary Fig. 13D) and our data showing no observable morphological or immunohistochemical changes in the markers examined following astrocytic-*Tgfb1* KO (Fig. 3 and Supplemental Figs. 6–8), RNA-seq data shows *Tgfb1*, *Tgfbr1*, *Tgfbr2*, and *Lrrc33* (a protein that is necessary for latent TGF-β1 ligand activation) are all significantly enriched in microglia compared to astrocytes (Supplementary Fig. 18, % mRNA levels in microglia vs astrocytes: *Tgfb1* = 500%, *Tgfbr1* = 9500%, *Tgfbr2* = 2600%, *Lrrcc33* = 500%). Instead, astrocytes express *Tgfb2*, which is absent in microglia, suggesting *Tgfb2* might have a potential role in astrocyte function (Supplementary Fig. 18 or Supplementary Data 2). We observed multiple compensatory mechanisms in response to MG-*Tgfb1* deletion: (1) an up-regulation of *Tgfb3* gene in microglia and no change of *Tgfb2* levels in astrocytes, (2) the upregulation of *Lrrc33* in microglia, a gene that has been demonstrated to be required in activating the latent TGF-β ligand[18], and (3) downregulation of *Smad7*, which is a negative

regulator of the TGF-β signaling pathway. Moreover, we observed a downregulation of *Smad3* and *Tgfbr1* but not *Tgfbr2* mRNA, suggesting a TGF-β signaling-dependent feedforward regulation of *Smad3* and *Tgfbr1* expression (Fig. 7B and Supplementary Fig. 18). Since our bulk RNAseq analysis used mixed female and male samples, we further validated several key differentially expressed genes in sorted microglia from an additional independent cohort of female and male WT and MG-*Tgfb1* iKO mice. qRT-PCR data from sorted microglia confirms that we do not observe a sex difference in all the examined genes (both upregulated and downregulated DEGs) in female or male *Tgfb1* iKO mice (Supplementary Fig. 19).

## MG-Tgfb1 iKO leads to cognitive, but not general motor deficits

We next investigated whether the DAM-associated and aging-associated microglia profile in the MG-*Tgfb1* iKO mice and the presence of reactive astrocytes in these mice could affect neurological function in young adult mice. Full dosage TAM was used in this experiment to achieve maximum changes in microglia and astrocytes in the adult brain (both female and male are used in the behavioral tests). A behavioral battery was used to examine general locomotion, motor coordination/learning, and cognitive function involving learning and memory. We first assessed voluntary movement in control and MG-*Tgfb1* iKO mice at 5 weeks after TAM injection using an automated open field locomotion tracking system and monitored mice for 23 h with free access to food and water (Omnitech Electronics INC, Columbus, OH). We did not observe any change in general locomotion in the *Cx3cr1*^CreER^*Tgfb1*^fl/fl^ + TAM animals compared to *Cx3cr1*^CreER^*Tgfb1*^wt/wt^ + TAM controls during the exploratory phase (1 h after naïve exposure to the chamber), or during the light or the dark cycle (Fig. 8K–N). Next, we carried out an acceleration rotarod test to evaluate motor coordination and motor learning. We specifically used a three-trial acceleration paradigm that starts at 1 rpm and increases to 35 rpm over the course of 5 min to evaluate their starting motor coordination, and how their performance improves over each trial. The *Cx3cr1*^CreER^*Tgfb1*^fl/fl^ + TAM animals did not show a difference in performance compared to the control *Cx3cr1*^CreER^*Tgfb1*^wt/wt^ + TAM group in the rotarod test (Fig. 8O), suggesting that motor coordination and motor learning are not affected in the MG-*Tgfb1* iKO mice at this time point after gene deletion. However, when we evaluated the cognitive function (spatial learning/memory) in control and MG-*Tgfb1* iKO mice using a 2-day Barnes Maze learning paradigm, *Cx3cr1*^CreER^*Tgfb1*^fl/fl^ TAM group showed an increase in latency to reach the escape hole and higher error trial numbers to locate the hole compared to the control mice (Fig. 8P–R), suggesting impaired spatial learning in the *Cx3cr1*^CreER^*Tgfb1*^fl/fl^ iKO mice. Importantly, *Cx3cr1*^CreER^*Tgfb1*^fl/fl^ mice that received vehicle treatment do not show any difference compared to control mice in any of the above behavioral tests (Fig. 8C–J),

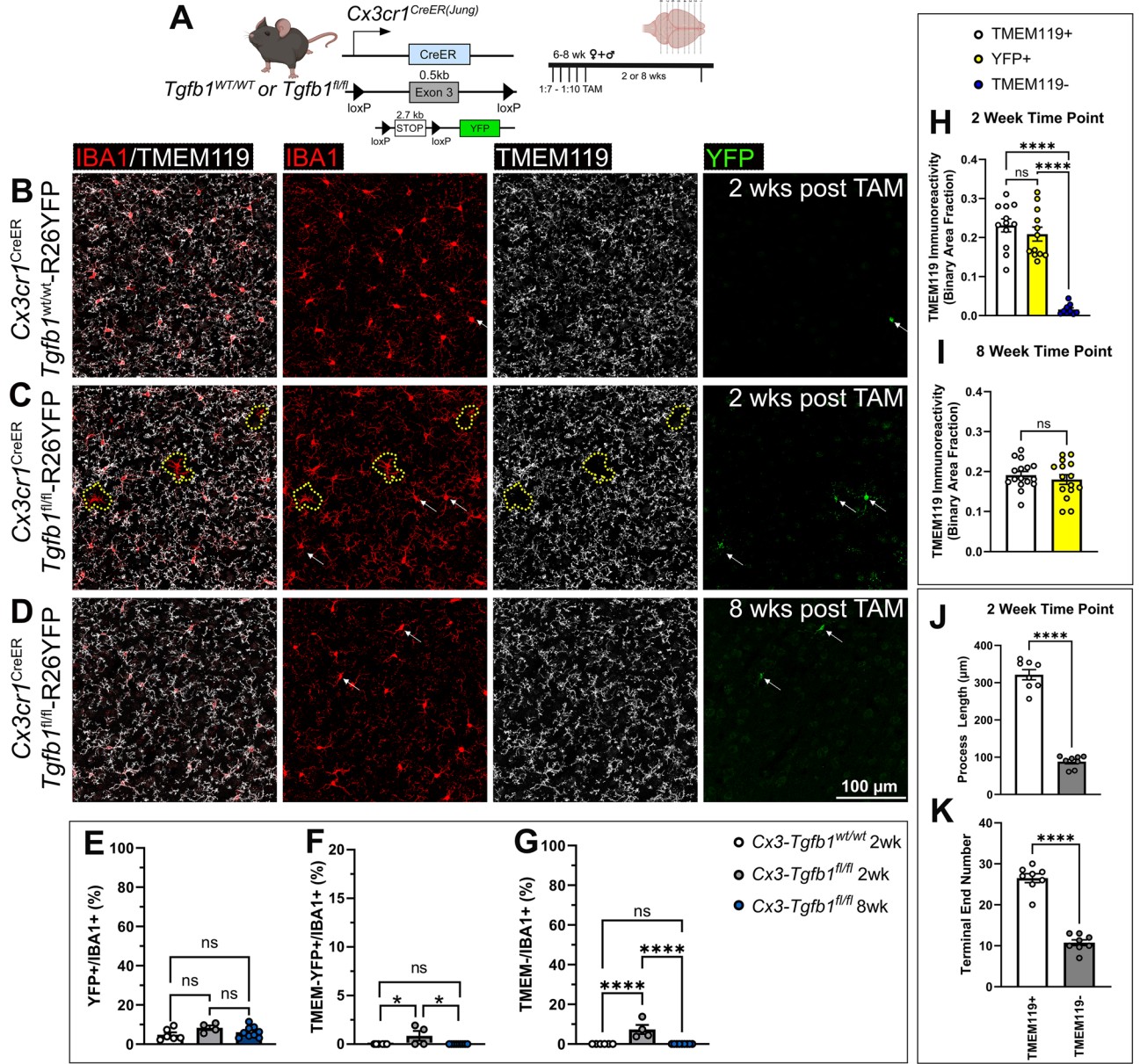

**Fig. 5 | Sparse-induced knockout of the *Tgfb1* gene in individual adult microglia supports an autocrine mechanism of microglial TGF-β ligand production and signaling regulation.** **A** Mouse model was used to induce *Tgfb1* KO in mosaic sparse individual microglia and an experimental timeline depicting a titrated dose of TAM. **B–D** Representative images showing IBA1, TMEM119, and YFP expression and co-localization in **B** control tissue at 2 weeks post TAM, **C** sparse iKO tissue at 2 weeks showing loss of TMEM119 expression in sparse individual microglia, and **D** absence of TMEM119−/IBA1+ parenchyma microglia in the sparse *Tgfb1* iKO brain at 8 weeks post TAM. The yellow dotted outline in **C** highlights singular microglia showing loss of homeostatic TMEM119 expression. White arrows highlight YFP+ cells showing no loss of homeostatic TMEM119 expression. Note that at this low dosage of TAM, the recombination of individual floxed alleles (R26-YFP reporter or the floxed *Tgfb1* gene) occurs independently of each other, therefore YFP+ cells could not track a sparse *Tgfb1* KO microglia, consistent with our recent study[32]. **E, F** Quantification of percentages of cell populations for **E** YFP+, **F** TMEM− YFP+,

and **G** TMEM− cells out of total IBA1+ cells at 2 and 8 weeks post sparse TAM administration (for **E–G**, *n* = 6, 4, and 9 for each group presented, *p = 0.0195 and 0.0123 for (**F**) and ****p < 0.0001, **E–G** analyzed by one way ANOVA, two-sided, Tukey's multiple comparison). **H, I** Quantification of TMEM119 expression at **H** 2 weeks post (*n* = 12, 12, and 9 for each group, ****p < 0.0001, one-way ANOVA, two-sided, Tukey's multiple comparisons) and **I** 8 weeks post (*n* = 16 and 15 for each group, not significant) low dose TAM administration from individual TMEM+ cells, TMEM− cells, and YFP+ cells. **J–K** Detailed morphological analysis of individual microglia in sparse iKO mice at 2 weeks post TAM characterizing **J** the total process length and **K** the total terminal end number of individual TMEM119+ or TMEM119− microglia (*n* = 8 for each group for **J** and **K**, ****p < 0.0001, **I–K** analyzed by Student's *t*-test, two-sided). Animals pooled from different cohorts of TAM treatment. Mean ± SE. Scale bar = 100 μm. **A** Created with BioRender.com and released under a Creative Commons Attribution-NonCommercial-NoDerivs 4.0 International license. Source data are provided as a source data file.

demonstrating that the behavioral deficits in cognitive function measured by Barnes Maze in the *Cx3cr1*^CreER^*Tgfb1*^fl/fl^ + TAM mice are specifically caused by TAM-induced deletion of the microglial-*Tgfb1* gene in these mice. These data support that microglia-derived TGF-β1 ligand is required in maintaining microglia homeostasis, astrocyte quiescence,

and normal cognitive function in the adult brain. We did not observe significant differences between female and male mice in either genotype.

Loss of TGF-β signaling during embryonic development is known to induce neuronal apoptosis[55], demyelination, and altered

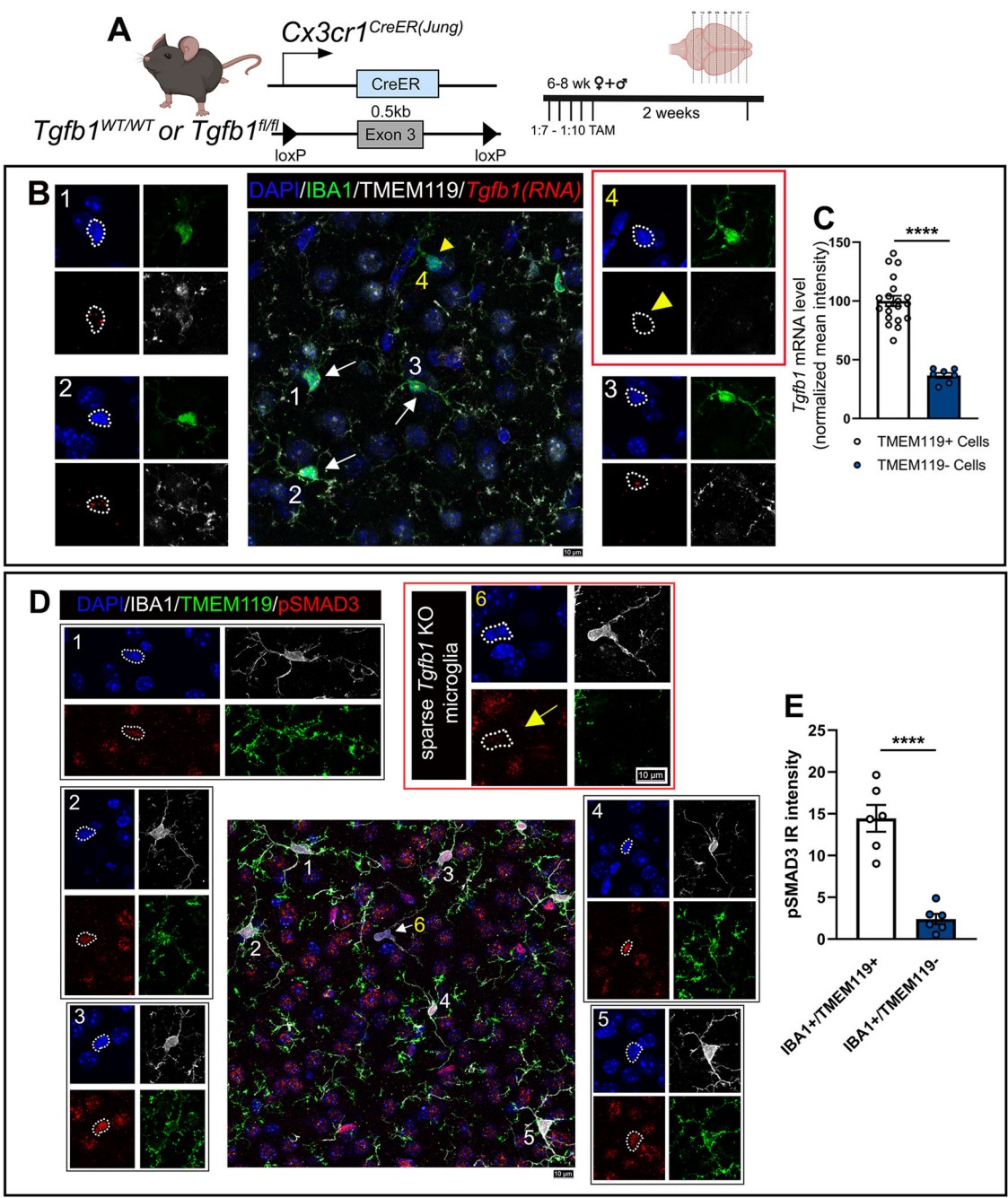

**Fig. 6 | In situ RNA-scope and IHC double labeling confirm loss of *Tgfb1* mRNA and downregulation of TGF-β downstream signaling (pSMAD3) in dyshomeostatic individual microglia in the sparse *Tgfb1* iKO model. A** Mouse model was used to examine sparse iKO in microglia and the experimental timeline with TAM dosage. **B** Representative image showing combined immunohistochemistry staining (for IBA1, TMEM119, and DAPI) and *Tgfb1* RNA-scope hybridization. (B1–3) Surrounding normal microglia showing TMEM119 expression and *Tgfb1* mRNA presence. (B4) A single microglia cell with loss of TMEM119 expression and loss of *Tgfb1* mRNA. White arrows were used to mark normal cells in the central panel. Yellow arrowhead is used to mark individual iKO microglia. Note that tissue treatment for RNAscope analysis makes the IHC condition less ideal for morphology evaluation than regular IHC staining, however, IBA1 and TMEM119 expression are still distinguishable for individual WT or iKO microglia. **C** Quantification of RNAscope signal intensity for *Tgfb1* probe in TMEM119+ and TMEM119− cells ($n = 20$ and 7 for control and TMEM119- group, ****$p < 0.0001$, Welch's *t*-test, two-sided). **D** Representative image showing co-immunohistochemical staining with DAPI, IBA1, TMEM119, and pSMAD3. (D1–5) Surrounding normal microglia showing TMEM119 expression and pSMAD3 immunostaining. (D6) A single microglia cell with loss of TMEM119 expression and loss of pSMAD3 labeling. The yellow arrow (microglia #6) marks the individual iKO microglia. **E** Quantification of pSMAD3 immunoreactive intensity in TMEM119+ and TMEM119− cells ($n = 6$ mice for each group, ****$p < 0.0001$, Student's *t*-test, two-sided). Scale bar = 10 μm. Mean ± SE. For additional representative images see Supplementary Fig. 12. **A** Created with BioRender.com and released under a Creative Commons Attribution-NonCommercial-NoDerivs 4.0 International license. Source data are provided as a source data file.

oligodendrocyte lineage as well as loss of somatostatin (SST)+ or Parvalbumin (PV)+ interneurons in neonatal mice[10]. We next investigated whether a similar phenotype is also observed in adult MG-*Tgfb1* iKO mice (TAM administration was given to young adult mice, aged

6–10-weeks-old). Consistent with the lack of major motor deficits in the adult MG-*Tgfb1* iKO mice, we did not observe any significant differences in the number of Oligodendrocyte transcription factor 2 (OLIGO2+), neuron-glial antigen 2 (NG2+) or CC1+ cells and myelin

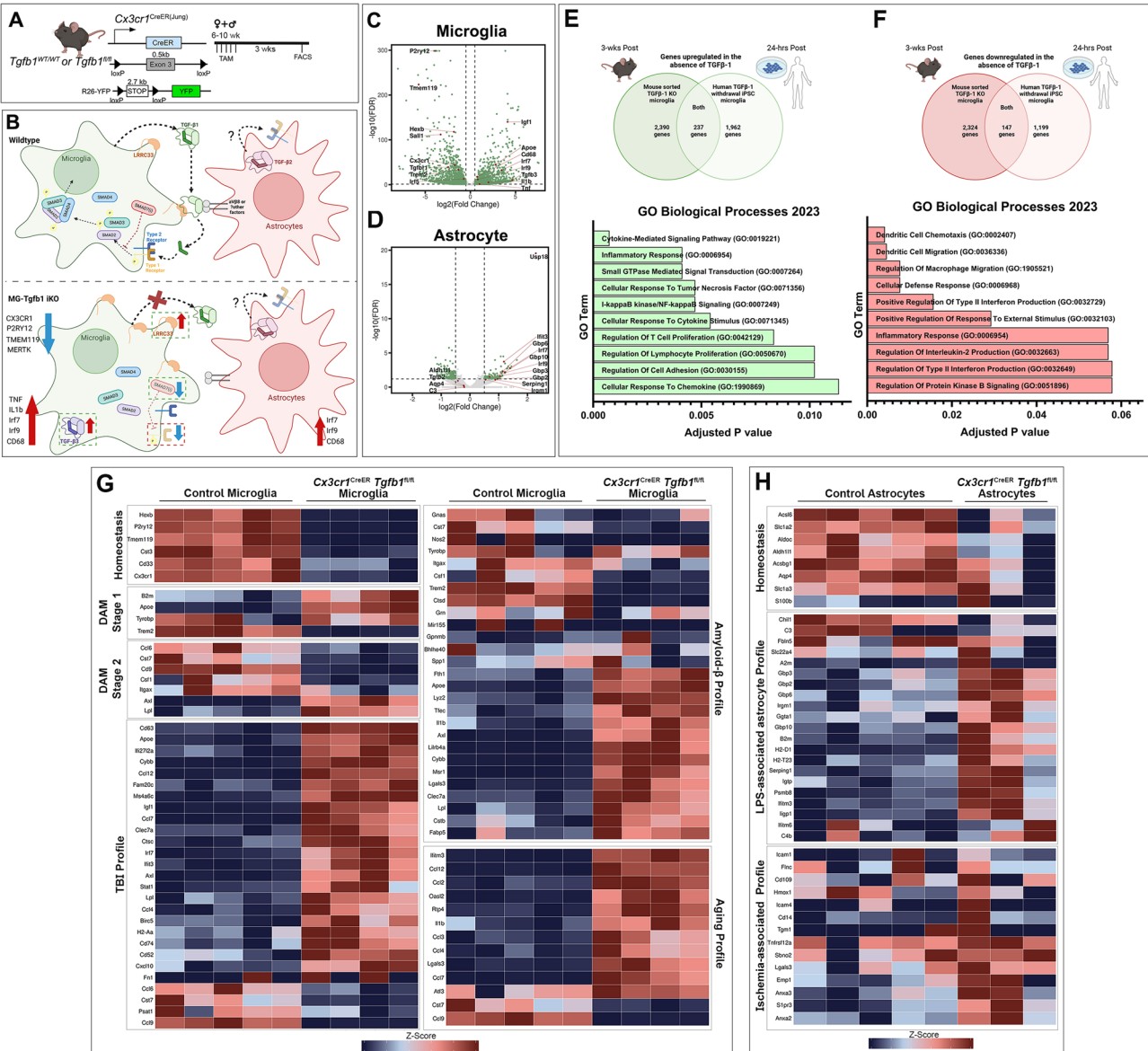

**Fig. 7 | Transcriptomic analysis of microglia and astrocyte cells sorted from *Cx3cr1*<sup>CreER(Jung)</sup> *Tgfb1* iKO mice.** **A** Mouse model was used to induce *Tgfb1* KO and *YFP* reporter in microglia. **B** Schematic model showing the summary of transcriptomic changes in microglia or astrocytes pertaining to both inflammatory responses and critical TGF-β signaling component genes, for individual gene list see supplemental information. **C**, **D** Volcano plot showing expression log fold changes in microglia or astrocytes comparing iKO vs Control mice. **E**, **F** Upregulated and downregulated genes common to this bulk RNA-seq data set and the sequencing results from Abud et al. (human microglia-like cells derived from iPSCs subjected to TGF-β withdrawal for 24 h) and gene ontology (GO) term analysis of overlapping genes from the two data sets. GO analysis was performed using the Enrichr online database. *p* Values were calculated using Fisher's exact test, and adjustments for

multiple comparisons were made using the Benjamini–Hochberg method. **G** Microglial differential gene expression observed across various gene sets including, homeostatic microglia genes[50–53], stage 1 and 2 disease-associated microglia (DAM) genes[51], injury exposed microglial (TBI)[53], amyloid beta exposed microglia[50,52,53], and aged microglia[52,53]. **H** Astrocytic differential gene expression was observed across different gene sets including, homeostatic astrocyte genes, LPS-associated, and ischemia-associated astrocytic genes. *Z*-scores were calculated and plotted to display differential gene expression[54]. The astrocyte sample that had an RIN < 8 was excluded from this analysis. **A**, **B**, **E**, and **F** Created with BioRender.com and released under a Creative Commons Attribution-NonCommercial-NoDerivs 4.0 International license. Source data are provided as a source data file.

basic proteins (MBP) levels in the cortical area (Supplementary Fig. 20). Similarly, there are no differences in the number or distribution of SST+ or PV+ interneurons (Supplementary Fig. 20) in the cortical layers of MG-*Tgfb1* iKO mice at 12 weeks post-TAM (a time point when microglia and astrocytes activation persist as shown in Fig. 1). Furthermore, the total NeuN+ neuronal population in all cortical layers at the cortical somatosensory region does not differ between control of MG-*Tgfb1* iKO mice at 12 weeks post TAM (Supplementary Fig. 20). These data suggest that, unlike the constitutive or global loss of *Tgfb1* ligand or receptors during the developmental stage[8,10], adult microglial

*Tgfb1* gene ablation has less effect on the adult oligodendrocyte lineages, the overall neuronal survival or the cortical interneuron population. This data also suggests that the observed cognitive deficits might be due to more subtle structural or functional changes in neurons in the iKO mice. Given the downregulation of multiple genes that are implicated in neuronal-microglia communications (P2RY12, CX3CR1)[56,57] and receptors that are required for phagocytosis function in microglia (P2RY12, MER proto-oncogene tyrosine kinase-MerTK and triggering receptor expressed on myeloid cells 2-TREM2)[58,59], we next hypothesized that synaptic pruning might be

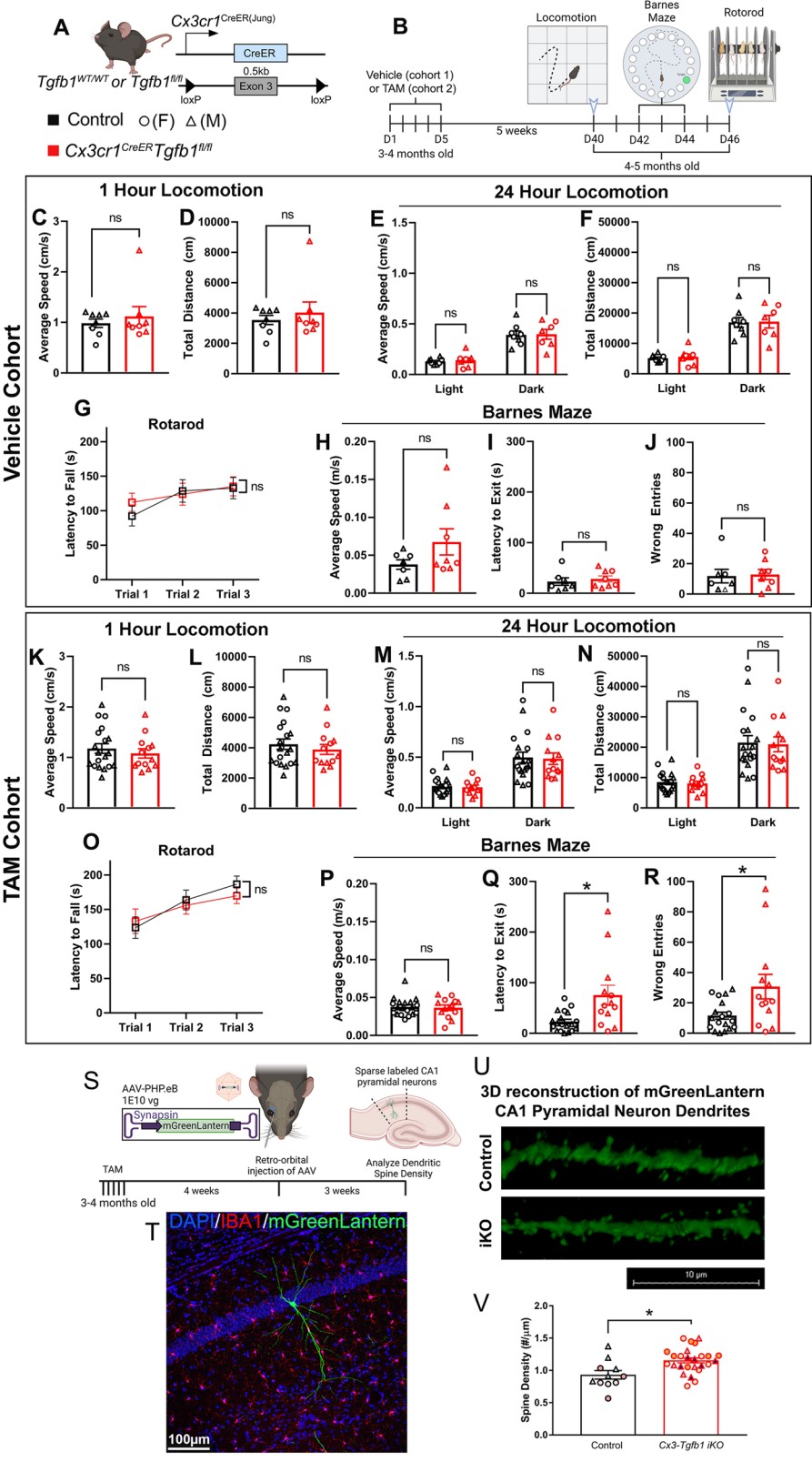

decreased in the MG-*Tgfb1* ligand knockout mice. To facilitate the measurement of dendritic spine density, neurons were sparsely labeled with the bright, monomeric fluorescent protein mGreen-Lantern (a bright monomeric fluorescent protein)[60] using a retro-orbitally delivered systemic Adeno-associated viruses (AAV) vector[61] (AAV-PHP.eB-hSyn-mGreenLantern-WPRE-pA, 1E10 vg/mouse) in WT or iKO mice (Fig. 8S). Given the known role of the hippocampus in

learning and memory and the observed cognitive deficits in the iKO mice, we analyzed the dendritic spine density of basal dendrites of the CA1 pyramidal neurons that are mGreenLatern positive (Fig. 8T). Our results show an increased spine density in the CA1 neurons of the MG-*Tgfb1* iKO mice (Fig. 8U, V), consistent with the observation of the downregulation of TREM2 and MerTK receptors in *Tgfb1* KO microglia.

**Fig. 8 | Behavioral assessment shows normal general motor function and motor learning but defective spatial learning and memory in young adult *Cx3cr1*[CreER]*Tgfb1*[fl/fl] iKO mice and increased dendritic spine density in hippocampal CA1 neurons. A** Mouse model was used to induce *Tgfb1* KO in microglia. **B** Experimental timeline, showing the order of behavioral measurements. **C–J** Behavioral measurements in vehicle-treated *Cx3cr1*[CreER]*Tgfb1*[wt/wt] or *Cx3cr1*[CreER]*Tgfb1*[fl/fl] mice showing open field test (OFT) of the first hour in locomotion chamber **C** average speed and **D** total distance traveled. **E, F** Average speed and total distance during the light and dark cycles in a 23-h period. **G** Accelerated rotarod learning test. **H–J** Barnes maze test showing **H** average speed during testing, **I** latency to locating the target hole, and **J** number of error trails before locating the target hole (*n* = 8 for each group, Student's *t*-test, two-sided). **K–R** Behavioral measurements from TAM-treated control and iKO mice showing open field test (OFT) of the first hour in locomotion chamber **K** average speed and **L** total distance traveled. **M, N** Average speed and total distance during the light and dark cycles in a 23-h period. **O** Accelerated rotarod learning test. **P–R** Barnes maze test showing

**P** average speed during testing, **Q** latency to locating the target hole, and **R** number of error trails before locating the target hole. (control *n* = 19, iKO *n* = 13) (ns = not significant, *\*p* = 0.0210 for panel **Q**, and *\*p* = 0.0398 for panel **R**, Unpaired *t*-test with Welch's correction, two-sided). **S** Experimental design and timeline for systemic AAV for neuronal labeling in MG-*Tgfb1* iKO mice. **T** Representative image of a sparse CA1 hippocampal neuron labeled with the AAV-PHP.eB syanpsin-mGreenLantern virus. **U** 3D reconstruction of mGreenLantern CA1 pyramidal basal neuron dendrites from control and iKO mice. **V** Quantification of spine density (*n* = 11 control and *n* = 26 individual dendritic segments pooled from *n* = 3 control mice and *n* = 4 iKO mice (dendritic segments from the same animal is color coated, circle indicates female, and triangle indicates male. Statistical analysis was carried out using the average from individual mice as a single *n*, *\*p* = 0.0316, Student's *t*-test, two-sided). Mean ± SE. Scale bars: 10 μm or 100 μm as indicated. **A**, **B**, and **S** Created with BioRender.com and released under a Creative Commons Attribution-NonCommercial-NoDerivs 4.0 International license. Source data are provided as a source data file.

## MG-Tgfb1 iKO repopulate dyshomeostasis microglia after MG ablation

Lastly, ablating and repopulation of microglia in dysregulated CNS have recently been suggested as a strategy to "reset" the inflammatory environment of the CNS[28,62]. However, since multiple studies have suggested an overall decreased TGF-β signaling in aging or disease brain[63–65], whether the compromised TGF-β signaling in microglia would affect the repopulated microglia under these contexts is not known. Next, we tested whether MG-*Tgfb1* or MG-*Alk5* deletion affects microglia repopulation in the adult brain following pharmacological microglia ablation (via the CSF1R inhibitor, PLX5622). Our data supports that while abolishment of TGF-β signaling does not prevent repopulation of microglia in the CNS after PLX5622 treatment, a homeostatic state is not reached in the repopulated microglia when TGF-β signaling is silenced in microglia via either microglia-specific ligand knockout (MG-*Tgfb1* iKO) or receptor knockout (MG-*Alk5* iKO). In the absence of microglia-derived TGF-β1 ligand or loss of ALK5 receptors, microglia overpopulate the brain (with an excess number compared to WT mice, Fig. 9A–D, H) and the repopulated microglia show activated morphology (Fig. 9E, F) and lack the homeostatic microglia signature gene expression (Fig. 9B–D). Importantly, the repopulation of dyshomeostatic *Tgfb1* or *Alk5* knockout microglia also leads to the activation of astrocytes (indicated by GFAP upregulation in astrocytes, Fig. 9G). This indicates that repopulation/resetting of the microglia population in a disease context where TGF-β signaling is diminished might lead to repopulation of non-homeostatic microglia, and may not be an ideal strategy.

## Discussion

Furthering our understanding of how TGF-β1 signaling is precisely regulated in the brain can provide important insight into microglia function during steady state and disease conditions. Our study addresses several gaps in our knowledge about CNS TGF-β ligand production and regulation and sheds light on how alteration of a single cytokine gene (*Tgfb1*) in microglia could causally contribute to cognitive deficits in young adult mice in the absence of brain injury or other disease-causing stressors.

Currently, the prevailing understanding of the source of TGF-β1 ligands in the CNS has been speculated to be coming from multiple cell types and that TGF-β ligands can be widely shared among different cell types[13–16,39,40,47,55,66–68]. Several reviews have proposed the sharing of TGF-β1 amongst all the cell types, despite not yet having a well-rounded experimental understanding of TGF-β1 ligand production and distribution[12–17]. Our data supports that microglia-produced TGF-β1 ligand is required for maintaining microglial homeostasis and subsequent astrocyte quiescence in the adult CNS as early as neonatal stages. We used multiple myeloid- or microglia-CreER drivers to rigorously investigate this phenotype. Two independent *Cx3cr1*[CreER]*Tgfb1*

iKO mouse lines both lead to a global loss of microglia homeostasis revealed by morphological changes and downregulation of homeostatic gene expression such as *Tmem119* and *P2ry12* without affecting serum or spleen levels of TGF-β ligand (demonstrated by both ELISA and FACS). We also show that astrocytic (via either inducible *Aldh1l1*[CreER] line or constitutive postnatal deletion via the *mGfap*[Cre] driver) or neuronal (via inducible *Camk2a*[CreER] line) deletion of the *Tgfb1* gene does not affect microglia morphology or expression of signature homeostatic microglia genes such as TMEM119 or P2RY12. Recent studies by us and others show that the *Cx3cr1*[CreER] mouse lines[31,45] recombine a portion of splenocyte macrophages even after the waiting period of >3 weeks, therefore, there is a possibility that the changes in microglia phenotype observed in the *Cx3cr1*[CreER]*Tgfb1* iKO mice could be due to parenchyma microglia population depletion and peripheral macrophage replacement in the brain. Alternatively, the activation of *Tgfb1* KO BAMs (which are also targeted by the *Cx3cr1*[CreER] lines) could subsequently activate the rest of the parenchyma microglia. However, results from the *P2ry12*[CreER(wt/mut)]*Tgfb1*[fl/fl] mice and *Tmem119*[CreER(wt/mut)]*Tgfb1*[fl/fl]R26-YFP mice also showed morphological and homeostatic microglial marker expression changes in the mosaic patches of YFP+ cells (indicating they were *P2ry12*+ or *Tmem119*+ parenchymal microglia at the time of TAM administration). Additionally, if peripheral monocytes (YFP−) are infiltrated into the brain, we expect to observe a decreased percentage of YFP+/IBA1+ cells in the brain. Instead, our data shows that total YFP+ and total IBA1+ cells in the brain both increased in the *Cx3cr1*[CreER]*Tgfb1* and homozygous *P2ry12*[CreER(mut/mut)]*Tgfb1* iKO mice while the percentage of YFP+/IBA1+ in total IBA1+ cells remained the same compared to control mice. This suggests that loss of TGF-β1 ligand in microglia does not lead to cell death of microglia or the replacement of microglia by peripheral monocytes but instead suggests increased proliferation of microglia in iKO mice, consistent with a previous study with *Tgfbr2* knockout[9] and supported by Ki67 gene upregulation in our iKO microglia (RNAseq). In summary, these data support that the parenchymal resident microglia are altering their phenotype in response to the deletion of the *Tgfb1* gene, rather than being replaced or indirectly altered by peripheral monocytes or macrophages.

Additionally, by using low TAM dosage to achieve sparsely mosaic *Tgfb1* KO in very few individual microglia, our data supports that not only do microglia produce their own ligand to regulate their quiescent state during adulthood, but they likely do so in an autocrine manner since *Tgfb1* gene deletion in sparsely distributed individual microglia leads to downregulation in TGF-β signaling (pSMAD3) and phenotypic changes in individual cells despite their surrounding WT microglia population. This data suggests that microglia regulate TGF-β signaling and related downstream pathways in a spatially precise manner which is consistent with the very low concentration of TGF-β1 ligand in brain tissue compared to spleen and serum levels. This mechanism is of

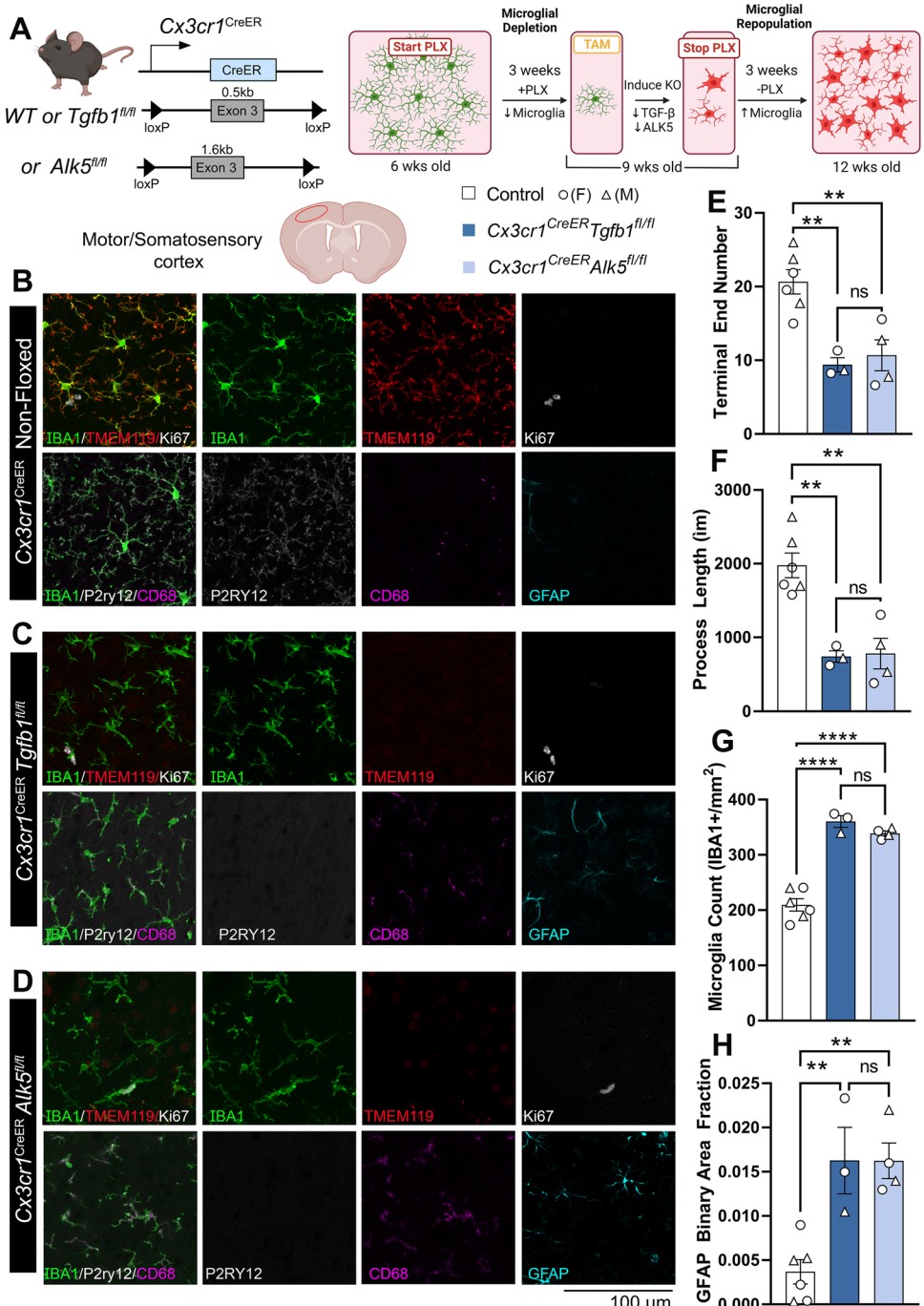

**Fig. 9 | TGF-β signaling via microglia-derived TGF-β1 ligand or ALK5-dependent signaling is required for the repopulation of homeostatic microglia after PLX5622 ablation. A** Mouse model for *Tgfb1* and *Alk5* iKO in microglia and experimental design. **B, C** Representative images of **B** control, **C** *Cx3cr1^CreER^Tgfb1^fl/fl^*, and **D** *Cx3cr1^CreER^Alk5^fl/fl^* mice showing immunostaining of IBA1, TMEM119, Ki67, P2RY12, CD68, and GFAP. Quantification of microglial morphology by **E** terminal end number (**$p = 0.0044$ and $p = 0.0053$) and **F** and process length (**$p = 0.0022$

and $p = 0.0014$). **G** Total microglia count (****$p < 0.0001$). **H** Quantification of astrocyte reactivity using GFAP immunoreactivity (**$p = 0.0057$ and $p = 0.0032$). **E**–**H**, $n = 6, 3, 4$ for each group, one-way ANOVA, two-sided, Tukey's multiple comparisons). Mean ± SE, each data point represents the average of a single animal. Scale bar = 100 μm. **A** Created with BioRender.com and released under a Creative Commons Attribution-NonCommercial-NoDerivs 4.0 International license. Source data are provided as a source data file.

particular importance in relation to disease or injury response since the glial activation cascade could be reliant on local fluctuating TGF-β1 levels. This model contrasts with the current prevailing model of shared TGF-β ligand production and signaling among many different cell types in the CNS[17]. We believe that our model helps to clarify one of the reasons for current controversy regarding the levels and role of TGF-β signaling in disease, aging, and Alzheimer's disease (AD), which

could be at least partially a result of the focus solely on serum or cerebrospinal fluid (CSF) TGF-β levels, which according to our results might not be biologically relevant to local microglia status. While there is some discrepancy in whether human *Tgfb1* gene expression also enriches in microglia in the human CNS (with several published data-sets supporting enriched Tgfb1 expression in microglia, one study showing differences in control vs AD subjects in microglial-*Tgfb1*

expression pattern[22–25]), our mouse MG-*Tgfb1* adult iKO transcriptomic analysis show many overlapping upregulated or downregulated DEGs (differentially expressed genes) with a human cultured microglial-like cell transcriptomic dataset after TGF-β withdrawal, supporting the potential relevance of our study in human cells. Importantly, TGF-β is synthesized as a latent form (L-TGF-β) whose activation requires the release of the mature c-terminal domain from non-covalently bound latency-associated peptide (LAP)[18,69,70]. One recent study also suggested a possible highly localized mechanism responsible for the release of the mature TGF-β1 ligand from the non-covalently bound prodomain (LAP) based on the coordinated molecular action of microglia-expressed LRRC33 (functioning as a LAP binding protein) and αVβ8 integrin[18], possibly expressed on other cell types such as astrocytes. Our model agrees with this recent study[13,69] that TGF-β ligand release is controlled at the single cell level by multiple precise mechanisms. The Qin et al. paper demonstrated that the activation of active TGF-β ligand from its latent prodomain-bound form is controlled by LRRC33 protein expression on the microglia membrane[13]. Our study now shows that not only is the unlocking of TGF-beta active ligand controlled by LRRC33 with single-cell precision (shown by Qin et al.), but also even at the production level, TGF-beta ligand produced by single microglia is required for the homeostasis of the same cell. This spatially precise model and mechanism demonstrates another distinct layer of the mechanism of this type of autocrine signaling and is an important addition to the mechanistic repertoire that cells use to refine extracellular signaling.

While this microglial autocrine mechanism appears to be the primary mechanism for TGF-β1 signaling in microglia during homeostasis, we observed that sparse individual KO microglia surrounded by WT microglia could potentially recover to homeostatic state (measured by morphology and expression of TMEM119 and P2RY12) at 8 weeks after the loss of native TGF-β1 ligand production. Note that on the populational level (with full TAM dosage and when most microglia are *Tgfb1* KO) even at 12 weeks, microglia in the *Cx3cr1*^CreER *Tgfb1* iKO mice still show morphological changes and decrease of TMEM119 and P2RY12 expression. This suggests that the milieu environment of surrounding WT microglia is able to "reset" the sparse individual *Tgfb1* KO microglia in the sparsely mosaic KO mice but not so efficiently when the majority of the microglia are KO cells that are activated. The mechanism for this recovery and to what extent the remaining "normal" microglia can help "reset" the mosaic-activated microglia is not clear and warrants further investigation in future studies. Our RNAseq data from the full dose TAM treated MG-*Tgfb1* iKO mice show that not *Tgfb2* but *Tgfb3* levels are upregulated in the MG-*Tgfb1* KO microglia (Supplementary Data 4), raising the interesting question of whether upregulated microglial TGF-β3 levels are able to compensate for the loss of TGF-β1 ligand in microglia. Previous studies have shown that *Tgfb1*− and *Tgfb3*-specific single KO mice have different phenotypes and that the swapping of the code sequence between Tgfb1 and Tgfb3 genes only leads to partial rescue of the phenotypes, suggesting non-overlapping functions of the two ligands, which is also supported by unique biophysical properties between the two ligands in recent studies[71]. Our results also suggest that upregulation of the TGF-β3 detected at 3 weeks post TAM in *Tgfb1*-KO microglia was not able to rescue the phenotype in microglia at up to 12 weeks post TAM in full dosage recombined mice. A recent study presented Cryogenic electron microscopy (Cryo-EM) structures which show that LRRC33 only presents L-TGF-β1 but not the -β2 or -β3 isoforms due to the differences of key residues on the growth factor domains[18]. This molecular selectivity offered by microglia-expressed LRRC33 could possibly explain why upregulated TGF-β3 expression in the TGF-β1 KO microglia could not compensate for the loss of TGF-β1 ligand and rescue the phenotype in microglia in full TAM dosage MG-*Tgfb1* iKO mice. RNA-seq data from sorted microglia and astrocytes from WT or MG-*Tgfb1* iKO brains also reveal interesting cell type-specific transcriptomic

regulation of the TGF-β signaling components in different cells during homeostasis or in response to the disturbance of TGF-β signaling. Specifically, we found that TGF-β1 is mainly enriched in microglia while TGF-β2 is enriched in astrocytes. Correspondingly, microglia express LRRC33 which preferentially presents TGF-β1 instead of TGF-β2 or -β3 for ligand activation. These patterns might explain why the deletion of microglial TGF-β1 but not astrocytic TGF-β1 leads to the observed phenotypes in both microglia and astrocytes. Loss of microglial TGF-β1 ligand also leads to downregulation of *Smad3* and *Tgfbr1* but not *Tgfbr2* in microglia. This suggests a feedforward regulation of TGF-β signaling on the expression of SMAD3 and TGF-βR1 facilitating further TGF-β signaling, which agrees with a recent study utilizing *Smad4* knockout mice[9]. Conversely, *Tgfb1* KO microglia also upregulate *Tgfb3* and *Nrros* (LRRC33) while downregulating the inhibitory *Smad7*, reflecting an attempt to compensate for the loss of TGF-β signaling in KO microglia. These gene expression changes reflect a highly dynamic regulation of this signaling pathway and support the precise spatially regulated autocrine mechanism in microglia. Consistently, none of these changes in the TGF-β signaling components are observed in astrocytes, indicating that the transcriptomic changes observed in astrocytes in the MG-*Tgfb1* iKO mice are likely not due to direct loss of TGF-β signaling in astrocytes. While *Alk5* deletion in astrocytes does not lead to GFAP upregulation in astrocytes or microglia morphological changes, it is not clear whether other transcriptomic or functional changes occur in the *Alk5* deleted astrocytes. Additionally, whether genetic deletion of *Tgfb2* in astrocytes would lead to activation of astrocytes and transcriptomic changes also warrants further investigation in future studies.

In the absence of the endogenous microglia-derived TGF-β1 ligand, microglia showed reduced ramification, decreased parenchymal homeostatic microglia signature gene expression, increased pro-inflammatory cytokine expression, and upregulation of interferon response genes. The expression profile of the *Tgfb1* iKO microglia aligned with disease-associated microglia (DAMs)[51,53], which have been described using both injury models (TBI)[53] and disease states (amyloid beta pathology)[50–53]. Additionally, these transcriptomic profile changes also corresponded with observed gene expression changes in aging microglia[52,53], suggesting that TGF-β1 signaling in microglia can provide vital insights into injury, neurodegenerative disease, and aging. The microglial *Tgfb1* iKO phenotype observed in our study is consistent with that described after constitutive *Tgfbr2* deletion in myeloid cells by the *Cx3cr1*^Cre promoter during development[10] and a *Sall1*^CreER driver in adult[9], which are much more severe than the phenotype observed when *Tgfbr2* was deleted at P30[10] or in a separate study at 2 months of age using the adult *Cx3cr1*^CreER *Tgfbr2* inducible mice[26]. Factors such as the dosage and route of TAM treatment, efficiency of recombination of the floxed genes, and whether cre-mediated recombination leads to the complete absence of the target protein, or a truncated protein can all contribute to the severity of the phenotypes. One specific potential caveat regarding the adult *Cx3cr1*^CreER *Tgfbr2*^fl/fl iKO study[11] is that KO microglia were sorted from *Cx3cr1*^CreER(+/wt) mice which are heterozygous for the *Cx3cr1* gene while control microglia are sorted from *Cx3cr1*^CreER(wt/wt) mice which has both alleles of the *Cx3cr1* gene. Heterozygosity of *Cx3cr1* has previously been reported to cause changes in gene expression or function in microglia[72–76] and therefore might introduce additional confounds to the data interpretation. Additionally, both previous studies[10,11] used the same *Tgfbr2* floxed mouse line[77] which has shown deletion of exons 2/3 does not alter the reading frame of the remaining exons leading to a truncated protein with normal serine/threonine kinase activity. The *Sall1*^CreER *Tgfbr2*^fl/fl study[9] used a different floxed *Tgfbr2* mouse line[78] which could potentially explain the differences in the three *Tgfbr2* KO studies. In our study, we utilized a *Tgfb1* floxed mouse model (065809-JAX with 0.5 kb of floxed region) which leads to a frameshift and results in the complete absence of the active TGF-β1 ligand. This

might explain the much more robust phenotypes in our adult iKO mice.

We also observed transcriptomic changes in astrocytes in the MG-*Tgfb1* ligand mice, featuring GFAP protein upregulation and increased interferon response genes. While GFAP is a pan-reactive marker, RNAseq data also shows upregulation of multiple previously reported disease-associated potentially detrimental astrocyte markers. Deletion of the *Tgfb1* gene in astrocytes via the constitutive *mGfap*[Cre] driver line or the inducible *Aldh1l1*[CreER] line did not induce morphological changes in microglia, nor upregulation GFAP expression in astrocytes, suggesting the reactivity of astrocytes in the MG-*Tgfb1* iKO mice is likely secondary to microglial profile change instead of direct loss of TGF-β signaling in astrocytes. This is consistent with the absence of the *Tgfb1* gene in astrocytes which instead express *Tgfb2*. Loss of TGF-β signaling in microglia leads to the upregulation of multiple pro-inflammatory cytokines which could in turn mediate the crosstalk between KO microglia and neighboring astrocytes. Our results in the *Alk5* receptor knockout specifically in microglia or astrocytes support a direct role of TGF-β signaling in microglia and subsequent changes in astrocytes due to crosstalk between the knockout microglia and surrounding astrocytes. Indeed, a detailed analysis of the spatial relationship of the activated microglia patches to the GFAP+ astrocyte distribution supports a close spatial correlation of the size and number of dyshomeostatic microglia to local patches of GFAP+ reactive astrocytes that could be mediated through diffusion of local cytokines. One such potential crosstalk can be through TNF signaling since it has been shown that TNF can promote disease-associated astrocyte activation and the TGF-β1 ligand knockout microglia shows an increase in TNFα in KO microglia[79]. Additional candidates for this crosstalk are also predicted using the ligand-receptor-target gene analysis and suggest that Integrin Subunit Beta 1 (*Itgb1*) and low-density lipoprotein receptor (*Ldlr*) are potential candidates for mediating microglial-dependent astrocyte reactivity that warrant further investigation in future studies.

Recently, enforced repopulation of adult microglia in diseased brains has been explored as a potential therapeutic strategy to "reset" homeostasis during neuroinflammation of the CNS[28,62]. TGF-β signaling-related genes are regulated during this process[80]. However, whether TGF-β signaling itself plays a role in adult microglia repopulation is not known. Our data show that after PLX5622 ablation of microglia, when TGF-β signaling is silenced in microglia with either the TGF-β1 ligand or the type 1 receptor *Alk5* knockout, the microglia population repopulates with excess numbers of microglia and dyshomeostatic microglia with activated morphology and decreased expression of microglia homeostatic signature markers and upregulation of CD68, accompanied by activation of astrocytes with GFAP expression. In pathological and disease contexts where TGF-β signaling has been reported to be diminished[63–65], the ablation and "resetting" strategy on the microglia population might not be an optimal strategy due to the repopulation of the dyshomeostatic microglia population.

Lastly, behavioral analysis in the *Cx3cr1*[CreER(Jung)]*Tgfb1* young adult mice shows that at 5 weeks following TAM treatment, there are significant deficits in the spatial learning and memory of MG-*Tgfb1* iKO mice without affecting the general locomotion function or motor learning at the time of assessment. This moderate cognitive phenotype without severe motor deficits in our young adult MG-*Tgfb1* iKO mice is an interesting contrast to the much more severe motor deficits and early life lethality in the constitutive gene knockout of *Lrrc33*–/– mice[81] and *Itgb8*–/– mice[82,83], and embryonic microglial deletion of *Tgfbr2* via the *Cx3cr1*[Cre] driver[10]. This difference suggests that loss of TGF-β signaling in microglia is more detrimental and causes more severe functional consequences during development vs in the adult CNS, a result that is also consistent with the difference in other cellular phenotypes (loss of myelination and cortical inhibitory interneuron) between embryonic vs adult KO mice. We did not observe any sex differences in our iKO mice using immunohistochemical, transcriptomic, or

behavioral analysis. Whether the abolishment of TGF-β signaling in microglia via ligand or receptor gene deletions could cause a more severe cellular or behavioral phenotype in aged or injury/diseased context in different sexes is also an interesting topic that can be further investigated in future studies. During the preparation of this paper, a recent study also reported a deficit in learning using a Morris Water Maze test in *Crybb1*[Cre]*Smad4* cKO mice[84]. The *Crybb1*[Cre] driver targets embryonic macrophages (with some off-target recombination in OPCs and neurons) and therefore the observed learning deficits could result from a deficit of neurons and projections during development[84]. However, our data show that adult microglia rely on self-derived TGF-β1 ligands to maintain homeostasis and TAM-induced deletion of the TGF-β1 ligand in adult microglia leads to learning deficits in adult mice, suggesting an ongoing reliance on microglia-derived TGF-β1 ligand and TGF-β1 signaling in microglia to maintain normal cognitive function in adulthood. Given the downregulation of multiple receptors such as P2RY12, TREM2, and MERTK which are involved in neuronal activity sensing and phagocytosis-mediated synaptic pruning, we examined the dendritic spine density in the CA1 pyramidal neurons. We chose to analyze CA1 pyramidal neurons because of their well-known roles in spatial memory and learning[85–88] which is relevant to the learning deficits we observed in the Barnes Maze. The AAV-PHP.eB Serotype and a low dose titer via systemic delivery were chosen to sparsely label neurons with minimal disturbance on microglia morphology or function which could be a confound using intracerebral injection of virus. Using this method, we observed increased dendritic spines in the CA1 pyramidal neurons of the MG-*Tgfb1* iKO mice, consistent with the downregulation of TREM2 and MERTK expression in KO microglia. Our results are further supported by a previous study showing increased dendritic spine density and functional deficits in TREM2 ko mice[89]. This data establishes a link regarding TGF-β signaling in microglia with TREM2 expression and neuronal function. However, it is possible that peripheral tissue macrophages could also rely on autocrine TGF-β1 ligand for their homeostasis, for example, gut macrophages[90], which could also potentially have some impact on the behavioral phenotypes. In summary, our result may have important implications for the role of microglial-TGF-β1 signaling in cognitive deficits observed during aging, neurodegenerative diseases, or after CNS injury. While constant basal TGF-β1 signaling is necessary for microglial homeostasis, TGF-β1 levels change with aging, injury, and disease[12,13,15,40]. Our results support that microglia dysregulation caused by the loss of TGF-β1 ligand results in transcriptomic features resembling DAMs and aged microglia. This could play a causal role in driving the cognitive deficits observed in disease conditions and targeting TGF-β signaling might be a potential therapeutic strategy to mitigate these deficits.

## Methods

### Experimental model and subject details

**Animals.** The University of Cincinnati (UC) Animal Care and Use Program (ACUP) encompasses Laboratory Animal Medical Services (LAMS, animal facilities) and the Institutional Animal Care and Use Committee (IACUC) office. All animal protocols were approved by the IACUC (animal protocol number: 21-03-02-01). Mice were housed in the animal facility of the University of Cincinnati on a 14-h light/10-h dark diurnal cycle. Food and water were provided *ad libitum*. The Cre-loxP recombination system was utilized to achieve cell-type specific constitutive or inducible knockout of the *Tgfb1* or *Alk5* gene. *Cx3cr1*[CreER(Jung)] (JAX: 020940[45]), *Cx3cr1*[CreER(Littman)] (JAX: 021160[31]), *P2ry12*[CreER] (JAX: 034727[47]), *Tmem119*[CreER] (JAX: 031820[46]), *Mgfap*[Cre] (JAX: 024098[91]), *Aldh1l1*[CreER] (JAX: 029655[41]), and *Camk2*[CreER] (JAX: 012362[43]) transgenic mouse lines in which the expression of Cre recombinase are under the control of the *Cx3cr1* (myeloid cell), *P2ry12* (Microglia), *Tmem119* (Microglia and peri-vesicular fibroblast), mouse *Gfap* (astrocytes and adult neural stem cells), *Aldh1l1* (astrocytes), and

*Camk2* (forebrain excitatory neurons) promoters, respectively were purchased from the Jackson Laboratory. These animals were crossed with floxed *Tgfb1* mouse line (JAX: 065809) or with *Alk5*flfl mice (JAX:028701). Among all the mouse cre driver lines used in this study, the *Mgfap*Cre mouse line is the only line that is non-inducible and has previously been shown to induce loxP-specific gene recombination at perinatal stages in mice which targets a large percentage of astrocytes (>90%) and a small percentage of cortical neurons (<1.3%) and some oligodendrocytes (<6% of total reporter positive cells)[42]. All the other Cre driver lines have the CreERT2[92]. For all experiments, mice were euthanized by administration of avertin followed by transcardial perfusion with either PB then 4% PFA (for IHC) or 1xHBSS (for flow cytometry, FACS, RNAseq, and qRT-PCR), followed by removal of essential organs (brain). Avertin 40× stock was prepared by mixing 10 g of 2,2,2-tribromoethanol (Sigma T48402) in 10 ml 2-methyl-2-butanol (Sigma 240486), then diluting 1/40 in sterile saline.

**TAM administration.** TAM injections were administered based on mouse body weight (BW) as described previously[93]. One hundred eighty mg/kg of BW was administered via oral gavage for five consecutive days for full dose induction. Young adult mice were treated at 6–10 weeks of age with TAM (180 mg/kg). TAM solution was formulated from 100 µl EtOH and 900 µl of sunflower seed oil diluted with 30 mg of TAM powder. For sparse recombination, mice received 3 days of TAM at th dosage of 1:10 (18 mg/kg BW, we noticed that 1:7 dilution also gave sparse labeled individual cells and similar results to 1:10. Each lab should test the titration of dilution in their own lab). Sparse *Tgfb1* KOs were generated using a 1:7–1:10 dilution of TAM in the vehicle (EtOH and sunflower seed oil) to achieve the desired dosage. Note that mice that receive the vehicle, diluted dosage (1:7–1:10), or full dosage of TAM should be housed separately to prevent TAM cross-contamination between different groups. For neonatal TAM treatment, pups are treated on P3–P5 daily by directly injecting TAM solution into the stomach (50 µg/pup/day) and harvested on day P18.

**Microglia ablation via PLX5622 administration.** Mice were treated with either the PLX5622 diet (AIN-76A rodent diet With 1200 PPM PLX5622, formulated by research diets with PLX5622 provided by Plexxikon) or the control diet (AIN-76A rodent diet, research diets, NJ). Animals had ad libitum access to the diet and water for the entirety of the study. For measuring *Tgfb1* mRNA levels after microglia ablation, C57bl6/J wildtype mice were treated with control or PLX5622 diet for 7 days. Brain tissue was harvested and processed for qRT-PCR as described previously[94]. For the ablation and repopulation experiment as illustrated in the experimental timeline, mice were given *ad libitum* access to the PLX5622 diet for 21 days and treated with TAM at the end of the PLX 5622 treatment. Mice were then returned to a standard chow diet for the remainder of the experiment to allow for repopulation.

**Sparse viral labeling of neurons and dendritic spine density analysis.** To label neurons, we used Gibson assembly to clone an AAV expression vector utilizing a human synapsin promoter (hSyn) for neuron-specific expression of the fluorescent protein mGreenLantern (pAAV-hSyn-mGreenLantern-WPRE-bGH-polyA). This construct was produced using the AAV-PHP.eB capsid by the Viral Vector Facility at Cincinnati Children's Hospital Medical Center. To avoid injury-related microglial reactivity, 1E10 vg of the viral construct was administered retro-orbitally in 90 µl of sterile saline. Four weeks after TAM administration, mice were anesthetized with isoflurane during the unilateral retro-orbital injection. Three weeks were allotted for viral transduction. Sparse labeled mGreenLantern positive CA1 pyramidal neurons are imaged at basal dendrites. For quantification of dendritic spines,

images were captured on a Leica Stellaris eight confocal microscope with a 63× objective and Z-series (Z-step of 0.2 um) with 2.5× digital zoom and 2048 × 2048 pixels. To assess spine density, basal dendrites from multiple CA1 pyramidal neurons are analyzed for each animal, and the spine number/length of each dendritic segment is averaged for each animal and used as $n = 1$ for statistical analysis.

**qRT-PCR.** RNA was isolated from the cortex using the RNAqueous-Micro Total RNA isolation kit (AM1931, ThermoFisher Scientific). cDNA was then generated using superscript III reverse transcriptase (18080044, ThermoFisher Scientific) or iScript cDNA synthesis kit (1708890, BioRad). The cDNA was then used for qRT-PCR using probes for *Hmbs1* (hydroxymethylbilane synthase), *Hprt1* (hypoxanthine phosphoribosyltransferase 1), *Iba1*, *Tgfb1*, *Alk5*, *Tgfbr2*, *Sall1*, *Glast*, *Glt*1, and *Atp1b2*. cDNA levels were quantified using a Roche Light Cycler II 480. Quantification of qRT-PCR values was normalized using the housekeeping gene Hmbs1 CT value, which did not change between groups after manipulation to account for potential variability in cDNA preparations.

**ELISA.** For ELISA analysis, tissue was collected after perfusion with phosphate buffer solution and flash frozen in cold isopropyl alcohol. Mouse serum is collected by clotted blood without any anticoagulant for 30 min followed by centrifugation at 1500 g for 10 min at 4 °C. Serum is collected from the supernatant and frozen at −80°C. The tissue was sectioned with a cryostat to punch 2 mm punches of tissue. Tissue was placed in RIPA buffer then homogenized using sonication at 30% amplitude, for 3-s pulses with 2-s pauses. The BCA method was used to determine the total protein concentration in the samples and Quantikine ELISA Human TGF-β1 kit (R&D Systems, Minneapolis, MN) was used to analyze TGF-β1 ligand levels following the instruction from the manufacturer.

**Tissue collection for flow cytometry or FACS.** Mice were transcardially perfused with cold 1× HBSS for 2–3 min. The brains and spleens were extracted and mechanically dissociated with a scalpel before using the papain dissociation kit (9001-73-4, Worthington Biochemical Corporation). For spleens, following dissociation, red blood cells were lysed using ammonium chloride. For the brains, once dissociated, cells were suspended in a 37% percoll solution and spun at 800 g for 20 min to remove excess myelin and debris. The cells were collected, washed, and resuspended in FACS buffer containing PBS with 1% (*v/v*) fetal bovine serum and 0.1% (*w/v*) NaN3 (Sigma), and counted. The number of cell subpopulations in the CNS was determined by multiplying the percentage of lineage marker-positive cells by the total number of mononuclear cells isolated from the brain. Transcriptional and translational inhibitors actinomycin, anisomycin, and typtolide were used to prevent activation of microglia during the preparation of tissues as was previously described by Marsh et al. [95]. Inhibitors were added to the dissection solution, and the papain enzyme cocktail from the Worthington kit.

**Flow cytometry analysis of TGF-β1 expression.** To carry out flow cytometry analysis, the Fc receptors were initially blocked using anti-mouse CD16/32 (0.25 µg; ThermoFisher) for 15 min at 4 °C. Cells were then washed with FACS buffer and stained for the surface marker for 30 min at 4 °C using the specified antibodies. These antibodies included: CD45 (clone 30-F11), CD11b (clone M1/70), and TGF-β1 (clone TW7-16B4) (all from Biolegend). Cells were then washed with PBS and viability staining was performed using the LIVE/DEAD fixable dead cell stain kit (Invitrogen). Following viability staining, cells were washed with PBS and resuspended in FACS buffer for flow cytometry analysis. Cells were acquired on a BD Canto II and analyzed using FlowJo X software (vX10). As controls, fluorescence minus one (FMOs) was used

to place the gates for analysis. For flow cytometry analysis, cells were first gated according to FSC-SSC, then restricted to single cells and live cells. Myeloid cells were identified as CD45+ CD11b+.

**FACS of microglia and astrocytes for qRT-PCR and RNA-seq.** Gating was determined using the yellow fluorescent protein expressed by *Cx3cr1*[CreER]-R26-YFP for CNS myeloid cell collection, and ASCA-2 APC conjugated antibody (130-117-535, Miltenyi Biotec) for astrocytes. Any double-positive cells were excluded from the gating to improve the purity of the samples.

**Bulk RNA-sequencing.** Non-directional RNA-seq was performed by the genomics, epigenomics, and sequencing core at the University of Cincinnati. To summarize, the quality of total RNA was QC analyzed by Bioanalyzer (Agilent, Santa Clara, CA). About 100 pg total RNA was used as input for cDNA amplification using NEBNext single cell/low input RNA Library Prep Kit (NEB) under PCR cycle number 15. After Bioanalyzer QC, 20 ng cDNA was used for library construction under PCR cycle number 6. After library QC and quantification via Qubit quantification (ThermoFisher, Waltham, MA), individually indexed libraries were proportionally pooled and sequenced using NextSeq 2000 Sequencer (Illumina, San Diego, CA) under the sequencing setting of PE 2 × 61 bp to generate about 60 M reads. Once the sequencing was completed, fastq files were generated via Illumina BaseSpace Sequence Hub.

**RNA-sequencing analysis.** RNA-seq reads with adapter sequences or bad-quality segments were trimmed using Trim Galore! v0.4.2[96] and cutadapt v1.9.1[97].The trimmed reads were aligned to the reference mouse genome version mm10 with STAR v2.6.1e[98]. Duplicated aligned reads were removed using Sambamba v0.6.8[99].Gene-level expression was assessed by counting features for each gene, as defined in the NCBI's RefSeq database[100]. Read counting was done using feature-Counts v1.6.2 from the Rsubread package[101]. Raw counts were normalized as transcripts per million (TPM). Differential gene expressions between groups of samples were assessed with R package DESeq2 v1.26.0[102]. Gene list and log2 fold changes are used for GSEA[103,104] analysis using the GO pathway dataset. Plots were generated using the ggplot2[105] package and base graphics in R. The PCA analysis comparing astrocytes from control samples and astrocytes from MG-*Tgfb1* iKO mice shows one astrocyte sample from iKO mice diverge from other iKO samples and this sample had a lower RNA integrity number (below 8 while all other samples have RIN of >8), suggesting partial RNA degradation. We included this sample in the PCA plot, general DEG heatmap, and volcano plot, however, this sample was excluded for characterizing the astrocytic activation profile. NicheNet package v1.1.1 in R v4.0.2 was implemented to infer ligand-receptor interactions using 1754 and 100 differentially expressed genes (FC ≥ |1.5| and FDR < 0.5) from microglia and astrocytes, respectively. The curated ligand-receptor interactions from the NicheNet database were used as a reference.

**RNAseq dataset comparisons method.** To generate a list of upregulated and downregulated genes, genes from the DEG analysis were first filtered by adjusted *p* value (<0.1 for the mouse-sorted microglia dataset and <0.05 for the iMGL dataset). From these genes, upregulated and downregulated genes were sorted based on log2-fold change. To identify genes common to both the mouse-sorted microglia dataset and the human iMGL dataset, human gene names were first converted to orthologous mouse gene names using Ensembl's Biomart. After the conversion of the human gene names to mouse orthologs, filtering for genes either upregulated in both datasets or downregulated in both datasets revealed 237 upregulated and 147 downregulated genes. GO analysis was conducted on the upregulated and downregulated gene lists using Enrichr.

**Immunohistochemistry.** Mice were perfused with 4% PFA and drop-fixed overnight before being transferred to 20% sucrose, followed by 30% sucrose once the tissue had sunk. Then tissue was sectioned on a cryostat in 30 μm thickness and subjected to IHC as previously described[106]. Antibodies for GFP (1:1000, Invitrogen or 1:500, Aves), IBA1 (1:500, Abcam or 1:1000, Wako), P2RY12 (1:500, Biolegend), TMEM119 (1:2000, GeneTex), GFAP (1:1000, Sigma), NEUN (1:1000, Biolegend), CD68 (1:4000, BioRad), MBP (1:250, BioRad), CC1 (1:250, Millipore Sigma), Olig2 (1:500, RND Systems), NG2 (1:250, Millipore), Parvalbumin (1:1000, Swant), Somatostatin (1:250, Santa Cruz), Ki67 (1:300, Invitrogen) and pSMAD3 (1:150, Abcam) were used. Tissue was blocked for 1 h at room temperature (RT) in 4%BSA/0.3% Triton-X100, then incubated overnight at 4 °C in primary antibody. Tissue was then incubated for 2–3 h in appropriate secondary antibodies conjugated with Alexa fluorescence 488, 555, 647, or 790.

**RNA-scope.** To fluorescently label RNA, RNA-scope was employed using the ACD RNA-scope Multiplex v.2 kit. Samples for RNA-seq were perfused with cold PB and the brain was dissected out and drop fixed in 4% PFA for 7 h before transferring to 20% and 30% sucrose. Brains were sectioned on a Leica Cryostat at the thickness of 16um and directly mounted onto superfrost plus glass slides. RNAscope hybridization steps were carried out following the instructions from the manufacturer. A *Tgfb1* probe was used (443571, ACD Biosciences) to label ligand RNA and an ALK5 probe was used (406201, ACD Biosciences) to label the TGF-β1 type I receptor. Immunohistochemistry was carried out after the RNAscope hybridization as described above to identify different cell types or expressions of different homeostatic microglia markers. Confocal images were obtained for IHC and RNA-scope/IHC samples at the University of Cincinnati Imaging Core utilizing a Confocal microscope. 3D reconstruction of the *Z*-stacks was performed in LAS X software. Imaging was also analyzed using the Neurolucida image analysis program (MBF Bioscience) for detailed morphological analysis.

**Image analysis.** NIS elements were used for microglial morphology quantification. For cortical regions, we used bregma coordinates: AP ~ +0.5 mm. For hippocampal regions, we used bregma coordinates: AP ~−2mm. A trace function was utilized for measuring the microglia process length (μm) and a counting function was utilized to quantify the number of terminal ends for each microglial process. Reactive astrocytes were quantified by measuring the area of GFAP immunoreactive astrocytes in the field via thresholding against the background. CD68 quantification was accomplished through the image processing package of ImageJ, Fiji. *Z*-stacks from confocal imaging were merged into a max projection on the Leica Application Suite X (LASX), and then exported to Fiji. The threshold function was utilized on the fluorescent staining of CD68 against the background, and averages were taken of the mean area. On average 4–6 images from 3–4 brain sections at similar brain regions were analyzed per mouse and the value from multiple images was averaged for each mouse which is used as a single data point in statistical analysis. Neurolucida (MBF Bioscience) was utilized for 3D microglial reconstructions. Microglia cell bodies were constructed by tracing the outer perimeter with the cell body trace function each time a primary microglia process branched out. Then each process was traced with the process tracer function. For pSMAD3 quantification, ImageJ was used to identify IBA1+ and IBA1- nuclei, the nuclei were then traced, and the median fluorescent intensity was measured. Multiple microglia (7–10 microglia) were analyzed from randomly sampled multiple images in individual mice and the average from multiple cells was used as a single data point in statistical analysis.

**Behavioral assays**
**Locomotion.** To measure general locomotion function, an automated 42 × 42 × 31 cm Plexiglas open box for 23 h with ad libitum food and

water (on the lid) and fresh bedding on the chamber of the floor (Omnitech Electronics INC, Columbus, OH) with laser sensors were used to monitor animal behavior. Animals were housed in a dual chamber, allowing for scent exchange but not physical interaction. Lighting was set to mimic that of the animals' normal housing rooms, on a 14/10 h light/dark cycle. Data was automatically scored using Fusion (Accuscan, Columbus, OH, USA).

**Rotarod.** Using the Roto-rod Series 8 apparatus (IITC Life Science Inc., Woodland Hills, CA) mice were given an acceleration paradigm. The rod rotation started at 1 RPM and reached 30 RPM by the end of the 5-min session. The test concluded when the animal fell from the rod but was not returned to the home cage until all animals finished the test. Animals performed three trials and were allotted at least 15 min in their home cage before starting their next trial. Seventy percent ethanol was used to clean the apparatus in between animals.

**Barnes maze.** Barnes maze apparatus (Stoelting Company, Wood Dale, IL) was designed as a gray circular platform 91 cm in diameter, and 90 cm in height with 20, 5 cm diameter holes equally distributed around the edge of the platform. Of the 20 holes, 19 had 2 cm deep gray trays beneath, with one of the holes having a 5 cm deep escape box. A short challenging Barnes maze learning paradigm was used to assess spatial learning and memory. Animals were given two training sessions, 4 h apart, with each training session ending when the animal located and fully entered the escape box. LED lights, a heat lamp, and a fan were used to motivate escape behavior. The following day (24 h later) a test session occurred, with the test ending when the animal fully entered the escape box. In-between animals the maze was sanitized using 70% ethanol. Data was automatically scored using AnyMaze (Stoelting Company, Wood Dale, IL).

### Statistical analysis
All studies were analyzed using SigmaPlot. Results are expressed by mean ± SEM of the indicated number of experiments. Statistical analysis was performed using the Student's $t$-test, and one- or two-way analysis of variance (ANOVA), as appropriate, with Tukey post hoc tests. A $p$ value equal to or less than 0.05 was considered significant.). Graphs were made in GraphPad Prism and some portions of figures were generated with Biorender.com.

## Data availability
The RNA-seq data generated in this study have been deposited in the GEO database under accession code GEO: GSE236032 (https://www. ncbi.nlm.nih.gov/geo/query/acc.cgi?acc=GPL30172). Other RNA-seq data presented, but not generated, in this study include: (1) GEO: GSE89189 (generation of human microglia-like cells to study neurological disease Fig. 7), (2) https://singlecell.broadinstitute.org/single_ cell/study/SCP1879/synucleinopathy-associated-astrocytes?genes= Tgfb1&tab=distribution#study-visualize (mouse single-cell RNA-seq database S Fig. 1), (3) https://brainrnaseq.org/ (mouse and human RNAseq database S Fig. 1), and (4) https://celltypes.brain-map.org/ rnaseq/human_ctx_smart%20seq?selectedVisualization=Scatter +Plot&colorByFeature=Gene+Expression&colorByFeatureValue=GAD1 (human single cell RNAseq database S Fig. 1). Microscopy data and behavioral test data reported in this paper will be shared by the lead contact upon request. Source data are provided with this paper. Any additional information required to reanalyze the data reported in this paper is available from the lead contact upon request. Source data are provided with this paper.

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

## Acknowledgments

Y.L. is supported by NIH grants (R01NS125074, R01AG083164, and R21NS127177). A.B. is supported by NIH 1F31NS125930. I.I. is supported by NIH R35GM146890. K.W. is supported by NIH F31 NS129204. J.E.R. is

supported by a Cincinnati Children's Research Foundation Trustee Award and a Simons Foundation Autism Research Initiative (SFARI) Bridge to Independence Award (663007). We thank Chet Closson and the University of Cincinnati live imaging core (supported by NIH-S10OD030402) for technical support. We thank Brendan Chestnut for assistance with vector cloning, as well as Thouwa Samake and the Vector Production Facility at Cincinnati Children's Hospital Medical Center for AAV production. We also thank Dr. Xiang Zhang and the Genomics, Epigenomics, and Sequencing Core at the University of Cincinnati for RNAseq analysis support. All figures were created using Biorender software.

## Author contributions

Y.L. conceptualized the study. Y.L., A.B., and E.W. designed the experiments. L.M. maintained all the mouse colonies and genotyped all mice in this study. A.B. and K.W. performed TAM injection and the immunohistochemistry staining with help from M.W., M.K.S., E.W., A.T., and J.D.P. A.B. performed all the behavioral analyses. A.B., M.W., and M.K.S. carried out microglia morphological analysis and astrocyte GFAP quantification. E.W. carried out the ELISA, microglia ablation experiment, and qRT-PCR analysis of the *Tgfb1* gene. A.B. and E.W. performed cell sorting of microglia and astrocytes with assistance from the Flow Cytometry core at CCHMC and prepared all RNA samples for RNAseq analysis. R.M.S.G. and J.E.R. designed and packaged the AAV-PHP.eB synapsin mGreen-Lantern virus at the CCHMC viral vector core. A.P. and K.M.R. assisted in the bioinformatics analysis of the RNAseq data in this study. A.A. and II carried out the flow cytometry analysis of *Tgfb1* expression in brain microglia and spleen myeloid cells. NG provided suggestions and consultants on experimental designs. A.B., E.W., N.H.G., and Y.L. drafted and revised the paper. All authors read, edited, and approved the final version of the manuscript.

## Competing interests

The authors declare no competing interests.
