## [Peer Review File · Nature Communications]

Adult Microglial TGF β 1 is required for microglia homeostasis via an autocrine mechanism to maintain cognitive function in miceREVIEWER COMMENTS

Reviewer #1 (Remarks to the Author):

The manuscript by Bedolla and colleagues present exciting data on TGFb1-mediated regulation of microglia homeostasis and activation. The authors have used a plethora of cell-specific Cre-lines to introduce CNS cell-specific inducible Tgfb1 ligand deletions in order to address which cellular source is essential for Tgfb1-mediated microglia homeostasis. The data presented in the current manuscript clearly demonstrate that microglia-derived Tgfb1 is necessary as evidenced using two individual Cx3cr1-Cre lines. Astrocytes (Aldh1l1 and Gfap-Cre) and excitatory neurons (Camk2-Cre) are not the main sources of Tgfb1 to keep microglia in a homeostatic and non-reactive state. Furthermore, the authors took advantage of the low recombination efficacy of Tmem119- and P2ry12-Cre lines to introduce Tgfb1-deletion in a mosaic pattern. These data prove that Tgfb1-production, secretion and induction of signal transduction seems to be restricted to the same cell. Neighbouring non-recombined cells cannot compensate the loss of Tgfb1 in individual cells. Moreover, using in-situ RNA-scope and IHC double labeling, the authors confirmed loss of Tgfb1 expression and downregulation of Smad3 phosphorylation in affected microglia.

The phenotypes described in the present study, define microglia with loss of Tgfb1- and Tgfb1-expression as being dyshomeostatic and reactive, presenting a transcriptional profile that resembles disease-associated microglia (DAMs). Finally, the authors demonstrate that depletion of microglia using an oral Csf1r-inhibitor (PLX5622), results in repopulation of dyshomeostatic and reactive microglia in microglia-specific Tgfb1- and Tgfb1-deficient mice. Interestingly, the authors show no motor phenotypes (which have been described in Tgfb1- and Smad4 microglia mutant mice) but a robust memory deficit which is accompanied by increased spine densities in granule hippocampal neurons.

The manuscript is well written and structured and has employed an impressive amount of distinct Cre-lines to define the cellular source of Tgfb1 in the CNS being essential for microglial homeostasis. The manuscript definitely contributes to increase our understanding of how Tgfb signalling regulates microglia functions and reactive states and further underlines the importance of Tgfb1 as therapeutic target.

The following points need to be addressed by the authors

1. Why was only Smad3 analyzed in the current study? Several studies have demonstrated that Smad2 also plays a major role in microglial Tgfb1 signalling.
2. The authors have used adult mice (6 weeks) to induce the Tgfb1-deletion in microglia. At this stage, homeostatic markers are robustly expressed in microglia. Have the authors considered to induce the deletion at early postnatal stages (P3/P5) to analyze whether microglial Tgfb1 is important to induce microglia maturation? Based on the data presented in the manuscript the conclusion would be that microglial Tgfb1 (and Tgfb1) is essential to keep the expression of homeostatic markers in adult mice. This should be commented and also discussed.
3. The authors speculate that the levels of YFP-expression indicates that no infiltration of peripheral monocytes occurs in their transgenic models. Have other monocyte (or macrophage) markers been used to confirm this?
4. Microglia-specific Tgfb1-KO mice have been shown to display disturbed myelination and loss of cortical inhibitory interneurons. However, when recombined at early postnatal stages. Have these well-described impairments been studied in the present adult mice? It might make a huge difference whether loss of Tgfb1 occurs at early postnatal stages or in adult mice. It would be nice to add this to the discussion part of the manuscript.
5. Have the authors seen different effects in male and female mice?
6. Although the important information to prove the efficacy of Tgfb1 deletion are presented in the supplementary part (which is again very impressive). It would be nice to add the information to show loss of Tgfb1 to the individual figures for the different Cre lines.

Reviewer #2 (Remarks to the Author):

In this manuscript Bedolla et al. utilize a wide range of Cre/lox mouse models to investigate TGFb-

mediated signaling cascades in glial cells of the brain, with a focus on the Tgfb1-Alk5 pathway in microglia. The authors mainly utilize the Cxc3cr1-iCre model to target select gene in resident brain microglia, showing that autocrine TGFb1 controls microglia and astrocyte homeostasis. Loss of Tgfb1 in microglial leads to alterations in cell morphologies, indicative of microgliosis, as well as upregulation of GFAP in astrocytes, suggesting astrogliosis. The authors also make the discovery that autocrine Tgfb1 signaling via Alk5 in microglia but no in astrocytes or neurons is involved in glial homeostasis. Defects in glial homeostasis are partially correlative with behavioral and cognitive impairment in mutant mice. Additional cell sorting and transcriptome sequencing experiments reveal potential Tgfb1-regulated pathways in astrocytes and microglia involved in cell homeostasis. The manuscript is well written and clearly presented. The results confirm prior global Tgfb1 and TGFb receptor gene deletion studies showing resulting microgliosis as well as reveal new microglia-specific roles for Tgfb1 in glial cell homeostasis. Most in vivo experiments are well controlled and supported by more than one Cre/lox model. The experimental analyses are statistically rigorous. However, the current manuscript does suffer from lack of mechanistic depth, especially related to how Tgfb1-Alk5 signaling selectively regulates microglial homeostasis and how Tgfb1 but not Alk5 is linked to astroglial homeostasis. As presented, there is a great deal of experimental data that leads to a somewhat incremental finding about the exact cellular source of Tgfb1 in the brain. This is important to know, but not greatly impactful, without inclusion of mechanistic studies of cell-cell signaling. In addition, better pathophysiological correlations between Tgfb1 signaling pathways and human brain diseases should be addressed.

Major criticisms

1. The authors have mined open-source mouse RNA sequencing databases to show that Tgfb1 is enriched in brain microglia. In the discussion section the authors speculate about how their results from their mouse models are potentially connected to human brain pathologies. However, these RNA expression patterns do not match well with the human BBB transcriptome database (https://twc-stanford.shinyapps.io/human_bbb/), as published in PMID: 35165441.
2. There is very careful and elegant usage of Cre/lox models to target the Tgfb1 signaling pathway in different glial and neuronal populations in the brain. FACS analysis and quantitation of TGFb1 in blood samples is included. However, PCR-based validation of Cre-mediated recombination of the different floxed genes is not sufficiently analyzed. The authors show in Fig. 6 using in situ RNAscope methods that Tgfb1 mRNA is reduced in microglia in mutant mouse brain tissue, but genomic PCR validation of sorted cells should be included.
3. The authors show that Tgfb1-Alk5 signaling in astrocytes is not involved in glial homeostasis. Tgfb1 can also signal via the type 1 receptor Alk1, but this possibility in microglia and/or astrocytes is not sufficiently explored.
4. The data in Figs. 8 and 9 should probably be merged. The synaptic density results in Fig. 9 are presumably included to explain the behavioral defects, but as presented they are somewhat tangential to the other findings. If the authors think there is a microglial-neuronal axis this should be explored mechanistically.
5. It is a bit confusing as to why there are such mild cognitive and motor deficits in the Tgfb1-/- mice (iCKO), given the extent of gliosis. The motor and cognitive deficits in Lrrc33-/- mice and Itgb8-/- (whole body or iCKO models), which have aberrant latent-Tgfb1 activation and signaling, are quite severe and lead to early lethality. This should be discussed in more detail.
6. More mechanistic data connecting astrocytes to microglia would add novelty to the findings. The authors have many powerful mouse models in hand as well as RNA sequencing data (Fig. 7) that should be used in additional experiments to delve deeper into Tgfb1-dependent cell-cell signaling mechanisms that control glial homeostasis in the brain. Seven potential ligand-receptor pairs have been identified from the RNA sequencing efforts and these should be explored further to add to the study, which is elegant but is lacking in mechanisms to account for the brain pathologies.

Minor points

1. A1/A2 terminology for astrocytes remains somewhat controversial and should probably be avoided (see PMID: 33589835).
2. The authors cite a relevant Lrrc33 reference (PMID: 29909984); however, the citations for the Itgb8-Tgfb connections are not totally relevant. A review article on Itgb8 summarizing genetic data linked to how Itgb8 activates Tgfb1 signaling would be more useful (PMID: 32540905).
3. A prior study which the authors cite (PMID: 14687548) has shown that Tgfb1-/- mice develop neurological deficits that correlate with brain microgliosis and neuronal apoptosis. The possibility of Tgfb1 deletion leading to neural cell death is not adequately explored in the current study. The

findings from this study, which is more than 20 years old, should be connected to the current data.

Reviewer #3 (Remarks to the Author):

In the article under review, Bedolla and colleagues demonstrate that TGFb1 is expressed mainly in microglia (not in astrocytes or neurons). Microglia also express its receptor what suggest that these cells are producers and responding to TGFb1. They showed using PLX5622 that microglia are a major source of TGFb1 in the brain. They were able to induce a loss of TGFb1 in microglia without affecting TGFb1 levels in the periphery. This impacted on microglial morphology, homeostatic markers and astrocytic reactivity. They demonstrate the TGFb1 has an autocrine role in microglia and wild type surrounding microglia could only rescue affected microglia if they were in low numbers. Also, after PLX5622 treatment, if TGFb1 is low, the phenotype of the microglia which is repopulating the CNS is not homeostatic what means that homeostatic microglia repopulation relies on TGFb1. Regarding behavioral outcome, they evaluated locomotion, learning and motor performance, as well as cognitive function. They showed that the loss of TGFb1 in microglia, impairs cognitive function in young adults. This is consistent with the loss of phagocytic markers and the increase of dendritic spines in the CA1 of the hippocampus. This work provides key information to understand important gaps in the field of microglial biology. The following points should be clarified to improve the manuscript:

- It should be clarified that only astrocytes gfap+ are being evaluated when analyzing how the loss of Alk5 affects microglial or astrocytic morphology/gene signature
- The sex of the animals used in the experiments should be mentioned. Microglial biology is extremely influenced by sex and this must be taken into account to discuss the results.
- Please add the age of the animals that were used.
- Information about the brain areas that were analyzed is needed. Particularly in the imaging section, please add information of which part of the cortex and hippocampus were used, specify the Bregma that was analyzed and clarify from which particular area the representative pictures shown were taken. This would be helpful to understand if the brain area influences the results.

Reviewer #4 (Remarks to the Author):

Reply to the reviewers' comments

We would like to thank the reviewers for their valuable comments and suggestions that have helped us substantially improve our manuscript further. Below is our point-by-point response to the reviewers' comments.

Reviewer #1 (Remarks to the Author):

The manuscript by Bedolla and colleagues present exciting data on TGF β 1-mediated regulation of microglia homeostasis and activation. The authors have used a plethora of cell-specific Cre-lines to introduce CNS cell-specific inducible Tgfb1 ligand deletions in order to address which cellular source is essential for Tgfb1-mediated microglia homeostasis. The data presented in the current manuscript clearly demonstrate that microglia-derived Tgfb1 is necessary as evidenced using two individual Cx3cr1-Cre lines. Astrocytes (Aldh1l1 and Gfap-Cre) and excitatory neurons (Camk2-Cre) are not the main sources of Tgfb1 to keep microglia in a homeostatic and non-reactive state. Furthermore, the authors took advantage of the low recombination efficacy of Tmem119- and P2ry12-Cre lines to introduce Tgfb1-deletion in a mosaic pattern. These data prove that Tgfb1-production, secretion and induction of signal transduction seems to be restricted to the same cell. Neighbouring non-recombined cells cannot compensate the loss of Tgfb1 in individual cells. Moreover, using in-situ RNA-scope and IHC double labeling, the authors confirmed loss of Tgfb1 expression and downregulation of Smad3 phosphorylation in affected microglia.

The phenotypes described in the present study, define microglia with loss of Tgfb1- and Tgfr1-expression as being dyshomeostatic and reactive, presenting a transcriptional profile that resembles disease-associated microglia (DAMs). Finally, the authors demonstrate that depletion of microglia using an oral Csf1r-inhibitor (PLX5622), results in repopulation of dyshomeostatic and reactive microglia in microglia-specific Tgfb1- and Tgfr1-deficient mice. Interestingly, the authors show no motor phenotypes (which have been described in Tgfr2- and Smad4 microglia mutant mice) but a robust memory deficit which is accompanied by increased spine densities in granule hippocampal neurons.

The manuscript is well written and structured and has employed an impressive amount of distinct Cre-lines to define the cellular source of Tgfb1 in the CNS being essential for microglial homeostasis. The manuscript definitely contributes to increase our understanding of how Tgfb signalling regulates microglia functions and reactive states and further underlines the importance of Tgfb1 as therapeutic target.

The following points need to be addressed by the authors

1. Why was only Smad3 analyzed in the current study? Several studies have demonstrated that Smad2 also plays a major role in microglial Tgfb1 signalling.

We chose pSMAD3 as the readout for TGF- β downstream signaling for two main reasons:

- SMAD3 is the DNA-binding Smad, therefore even if SMAD2 is phosphorylated it will be unable to bind DNA and function as a transcription activator in the absence of phosphorylated SMAD3.

- To the best of our knowledge, pSMAD3 is the only phosphorylated member of the SMAD complex for which antibodies are available that work well on IHC, which was published previously (PMID: 30846482). During the revision, we tested two different pSMAD2 antibodies that perform well in Western blot, but they did not generate a good signal in IHC. The pSMAD3 antibody, which was reported previously and used in this study, is the only phosphorylation-specific antibody that works for both Western blot and IHC.

*2. The authors have used adult mice (6 weeks) to induce the *Tgfb1*-deletion in microglia. At this stage, homeostatic are robustly expressed in microglia. Have the authors considered to induce the deletion at early postnatal stages (P3/P5) to analyze whether microglial *Tgfb1* is important to induce microglia maturation? Based on the data presented in the manuscript the conclusion would be that microglial *Tgfb1* (and *Tgfbr1*) is essential to keep the expression of homeostatic markers in adult mice. This should be commented and also discussed.*

- We performed inducible *Tgfb1* deletion in microglia at the early postnatal stage to address this question. We induced *Tgfb1* gene deletion in *Cx3cr1^{CreER}* mice by TAM administration on days P3 to P5 and harvested brain tissue at P18. Lack of pSMAD3 nuclear staining in IBA1+ cells confirmed abolishment of TGF- β signaling in neonatal microglia. Similar to adult mice, we observed microglial activation in this early postnatal stage, demonstrated by morphological changes in IBA1+ cells and a loss of TMEM119 and P2RY12 homeostatic markers in KO mice. Commentary on neonatal *Tgfb1* gene deletion in microglia has been added to the manuscript and discussion (Supplementary. Fig 5). This result is consistent with a recent preprint focusing on microglial-*Tgfb1* expression during the embryonic stage (PMID: 37790363) from Dr. Thomas Arnold's group at UCSF.

3. The authors speculate that the levels of YFP-expression indicates that no infiltration of peripheral monocytes occurs in their transgenic models. Have other monocyte (or macrophage) markers been used to confirm this?

- Unfortunately, peripheral monocyte/macrophage markers are an unreliable method to distinguish infiltrating monocytes from activated microglia under a non-physiological state (PMID:32042176). It has been demonstrated that microglia can upregulate markers typically associated with peripheral macrophages/monocytes when activated (PMID: 33261619). Specifically, it is known that loss of TGF- β signaling in microglia (via receptor knock-out) increases CD45 expression in microglia (PMID: 27776109). This is why we had to utilize YFP to preferentially label microglia before gene knockout, then use the ratio of YFP+/Iba+ as a readout of peripheral macrophage infiltration.

*4. Microglia-specific *Tgfbr2*-KO mice have been shown to display disturbed myelination and loss of cortical inhibitory interneurons. However, when recombined at early postnatal stages. Have these well-described impairments been studied in the present adult mice? It might make a huge difference whether loss of *Tgfb1* occurs at early postnatal stages or in adult mice. It would be nice add this to the discussion part of the manuscript.*

- As suggested, we have undertaken IHC staining for inhibitory neurons as well as oligodendrocyte markers in adult *Cx3cr1CreERTgfb1 iKO* mice at 12 weeks post TAM. We chose the 12-week timepoint to assess possible interneuron and oligodendrocyte

degeneration, as KO microglia still display the activated phenotype at 12 weeks and we wanted to evaluate the long-term effects on these populations. We observed no significant difference between WT and KO mice in cortical inhibitory interneuron numbers or oligodendrocyte markers (MBP, CC1, OLIGO2, NG2) or total NeuN+ populations in the cortex. This data is now added to Supplementary Fig 20 and in the results and discussion sections.

5. Have the authors seen different effects in male and female mice?

- Both male and female mice (in balanced numbers) are used in all of our experiments in all mouse models. We did not observe any apparent differences in the phenotype in any of our analyses. We did not have enough N to properly evaluate sex as a variable for all experiments. However, data points from male and female animals are now indicated as individual points (males are triangles, females are circles) in all main figures, and we did not detect any obvious trends based on sex differences.

6. Although the important informations to prove the efficacy of Tgfb1 deletion are presented in the supplementary part (which is again very impressive). It would be nice to add the informations to show loss of Tgfb1 to the individual figures for the different Cre lines.

- We now provide this data (as a panel to show gene deletion efficiency) in all of the microglia-related lines (Cx3cr1CreER, P2ry12CreER (heterozygous Cre allele and homozygous Cre alleles, and TMEM119CreER) in the corresponding figures (Fig1A', Fig 2A' and Fig 4A'). We have also reported some of the related information (regarding gene deletion efficiency in different lines in our recent Cell Reports paper, PMID 38217856).

As for the neuronal and astrocytic lines, unfortunately, due to the huge burden of maintaining so many different lines in this study, we no longer have these mouse lines. For the astrocytic KO line, we have frozen down primary cultured GFAP+ cells from WT and KO mice and we carried out PCR on genomic DNA to confirm efficient recombination of the floxed ALK5 alleles. We have now included this data in Figure 2H' for the astrocytic line.

For Camk2aCreER KO lines, we only have frozen thick (30µm) PFA-fixed tissue sections. As RNAscope does not work well for this tissue thickness, we have performed laser capture microdissection (LCM) in *Camk2a*^{CreER} iKO tissue followed by quantitative PCR on genomic DNA to detect levels of either the floxed genomic *Tgfb1* exon 3 (within the two loxP sites) or exon 5 (which is outside of the two loxP sites). qPCR reactions were carried out in multiplex using distinct primer/probe sets in the same reaction tube to ensure the same amount of gDNA template was amplified for the distinct exons. Due to the thickness of the tissue, it is difficult to isolate single cells that are free of contamination from other cell types, therefore clusters of NeuN+ or NeuN- cells were collected using LCM. About 20-30 cells that are enriched in NeuN- or NeuN+ cells were collected for this experiment. Using this method, we show a 50% decrease specifically in the floxed exon 3 in NeuN+ cells compared to NeuN- cells, supporting that *Tgfb1* is indeed deleted or decreased in neurons in this line. Combined with the extensive characterization of this Cre driver and many previous studies using this mouse line, we are confident about the neuronal deletion of *Tgfb1* gene

(which has a very short floxed cassette, about 0.5 kb). This information is provided below, and we can include it as a supplementary figure if needed (although recognizing the caveat that the LCM experiment is not ideal on 30µm-thick sections to obtain pure single cells).

Reviewer #2 (Remarks to the Author):

In this manuscript Bedolla et al. utilize a wide range of Cre/lox mouse models to investigate TGFb-mediated signaling cascades in glial cells of the brain, with a focus on the Tgfb1-Alk5 pathway in microglia. The authors mainly utilize the Cxcr3cr1-iCre model to target select gene in resident brain microglia, showing that autocrine TGFb1 controls microglia and astrocyte homeostasis. Loss of Tgfb1 in microglial leads to alterations in cell morphologies, indicative of microgliosis, as well as upregulation of GFAP in astrocytes, suggesting astrogliosis. The authors also make the discovery that autocrine Tgfb1 signaling via Alk5 in microglia but no in astrocytes or neurons is involved in glial homeostasis. Defects in glial homeostasis are partially correlative with behavioral and cognitive impairment in mutant mice. Additional cell sorting and transcriptome sequencing experiments reveal potential Tgfb1-regulated pathways in astrocytes and microglia involved in cell homeostasis. The manuscript is well written and clearly presented. The results confirm prior global Tgfb1 and TGFb receptor gene deletion studies showing resulting microgliosis as well as reveal new microglia-specific roles for Tgfb1 in glial cell homeostasis. Most in vivo experiments are well controlled and supported by more than one Cre/lox model. The experimental analyses are statistically rigorous. However, the current manuscript does suffer from lack of mechanistic depth, especially related to how Tgfb1-Alk5 signaling selectively regulates microglial homeostasis and how Tgfb1 but not Alk5 is linked to astroglial homeostasis. As presented, there is a great deal of

experimental data that leads to a somewhat incremental finding about the exact cellular source of Tgfb1 in the brain. This is important to know, but not greatly impactful, without inclusion of mechanistic studies of cell-cell signaling. In addition, better pathophysiological correlations between Tgfb1 signaling pathways and human brain diseases should be addressed.

Major criticisms

1. The authors have mined open-source mouse RNA sequencing databases to show that Tgfb1 is enriched in brain microglia. In the discussion section the authors speculate about how their results from their mouse models are potentially connected to human brain pathologies. However, these RNA expression patterns do not match well with the human BBB transcriptome database (https://twc-stanford.shinyapps.io/human_bbb/), as published in PMID: 35165441.

- Thank you for raising this important point. The above-mentioned study, interestingly, shows enrichment of Tgfb1 gene in microglia only in AD samples, not control microglia, which contradicts several other publicly available human single-cell RNAseq databases. We provided results from two widely accepted public scRNAseq platforms (BrainRNAseq.org and Allen Brain Institute scRNAseq database, Supplementary Fig 1C, D) for human brain scRNAseq analysis. Both of these databases show results supporting the enrichment of the Tgfb1 gene in human microglia. It is unclear why there is a discrepancy between the (PMID 35165441) study and multiple other publicly available human and mouse databases (supp. Fig 1). We cited all these papers and added a discussion on the comparison between human and mouse data. Our data in Figure 4 also supports the notion that there are common upregulated and downregulated DEGs in mouse vs human iMGs after TGF- β depletion.

2. There is very careful and elegant usage of Cre/lox models to target the Tgfb1 signaling pathway in different glial and neuronal populations in the brain. FACS analysis and quantitation of TGF β 1 in blood samples is included. However, PCR-based validation of Cre-mediated recombination of the different floxed genes is not sufficiently analyzed. The authors show in Fig. 6 using in situ RNAscope methods that Tgfb1 mRNA is reduced in microglia in mutant mouse brain tissue, but genomic PCR validation of sorted cells should be included.

- Besides RNAscope analysis, we carried out quantitative PCR analysis-based measurement of Tgfb1 mRNA levels in multiple mouse lines (including *Cx3cr1^{CreER}*, *TMEM119^{CreER}*, and *P2ry12^{CreER}* heterozygous Cre and homozygous Cre drivers) in sorted microglia. Part of this data was also published in our recent paper (PMID 38217856) and was considered as an acceptable means to evaluate gene deletion efficiency in conditional knockout lines. For the astrocyte lines, we have now added data showing complete genomic recombination of the floxed Alk5 exon 3 in primary GFAP+ cells collected from adult *mGfapCre- Alk5^{fl/fl}* mice (Fig 2H').

3. The authors show that Tgfb1-Alk5 signaling in astrocytes is not involved in glial homeostasis. Tgfb1 can also signal via the type 1 receptor Alk1, but this possibility in microglia and/or astrocytes is not sufficiently explored.

- It has been reported that TGF- β signaling through Alk1 requires Alk5 (PMID14580334). Alk1 can also mediate other functions that are Alk5 or TBR1 independent, suggesting that it can signal through non-TGF- β ligands (PMID: 17911384). Therefore, given the already large number of mouse models included in this study, we chose to focus on the main type 1 receptor for TGF- β signaling (ALK5).

4. The data in Figs. 8 and 9 should probably be merged. The synaptic density results in Fig. 9 are presumably included to explain the behavioral defects, but as presented they are somewhat tangential to the other findings. If the authors think there is a microglial-neuronal axis this should be explored mechanistically.

- We have merged the figures as suggested. The potential mechanisms of the possible microglial-neuronal axis are currently being investigated in our follow up study.

5. It is a bit confusing as to why there are such mild cognitive and motor deficits in the Tgfb1^{-/-} mice (iCKO), given the extent of gliosis. The motor and cognitive deficits in Lrrc33^{-/-} mice and Itgb8^{-/-} (whole body or iCKO models), which have aberrant latent-Tgfb1 activation and signaling, are quite severe and lead to early lethality. This should be discussed in more detail.

- We have expanded the discussion to address the possibility that loss of TGF- β signaling in microglia may be more detrimental and cause more severe deficits in the developing CNS versus the adult CNS.

6. More mechanistic data connecting astrocytes to microglia would add novelty to the findings. The authors have many powerful mouse models in hand as well as RNA sequencing data (Fig. 7) that should be used in additional experiments to delve deeper into Tgfb1-dependent cell-cell signaling mechanisms that control glial homeostasis in the brain. Seven potential ligand-receptor pairs have been identified from the RNA sequencing efforts and these should be explored further to add to the study, which is elegant but is lacking in mechanisms to account for the brain pathologies.

- We agree that these topics will be an excellent focus for future studies which is beyond the scope of the current study.

Minor points

1. A1/A2 terminology for astrocytes remains somewhat controversial and should probably be avoided (see PMID: 33589835).

- The astrocyte terminology has been updated to reflect the stimulus/stress used to induce the astrocytic profile in each specific dataset (in our case we compared it to the LPS-induced and ischemia-induced astrocyte transcriptomics).

2. The authors cite a relevant Lrrc33 reference (PMID: 29909984); however, the citations for the Itgb8-Tgfb connections are not totally relevant. A review article on Itgb8 summarizing genetic data linked to how Itgb8 activates Tgfb1 signaling would be more useful (PMID: 32540905).

- We have now included this reference as suggested.

3. A prior study which the authors cite (PMID: 14687548) has shown that *Tgfb1*^{-/-} mice develop neurological deficits that correlate with brain microgliosis and neuronal apoptosis. The possibility of *Tgfb1* deletion leading to neural cell death is not adequately explored in the current study. The findings from this study, which is more than 20 years old, should be connected to the current data.

- We have carried out NeuN⁺ neuron quantification in the cortex of adult *Cx3cr1*^{CreER}*Tgfb1* iKO mice at 12 weeks post TAM. We chose the 12-week timepoint to assess potential long-term neurodegeneration, as iKO microglia still display the activated phenotype at 12 weeks. At this timepoint, we observed no difference in neuron density between the iKO and control mice (Supplementary Fig 20). We believe that global and constitutive germline *Tgfb1* knockout may have more profound effects on neuronal survival than the adult *MG-Tgfb1* iKO. We have added this in the discussion section.

Reviewer #3 (Remarks to the Author):

In the article under review, Bedolla and colleagues demonstrate that TGFβ1 is expressed mainly in microglia (not in astrocytes or neurons). Microglia also express its receptor what suggest that these cells are producers and responding to TGFβ1. They showed using PLX5622 that microglia are a major source of TGFβ1 in the brain. They were able to induce a loss of TGFβ1 in microglia without affecting TGFβ1 levels in the periphery. This impacted on microglial morphology, homeostatic markers and astrocytic reactivity. They demonstrate the TGFβ1 has an autocrine role in microglia and wild type surrounding microglia could only rescue affected microglia if they were in low numbers. Also, after PLX5622 treatment, if TGFβ1 is low, the phenotype of the microglia which is repopulating the CNS is not homeostatic what means that homeostatic microglia repopulation relies on TGFβ1. Regarding behavioral outcome, they evaluated locomotion, learning and motor performance, as well as cognitive function. They showed that the loss of TGFβ1 in microglia, impairs cognitive function in young adults. This is consistent with the loss of phagocytic markers and the increase of dendritic spines in the CA1 of the hippocampus. This work provides key information to understand important gaps in the field of microglial biology. The following points should be clarified to improve the manuscript:

*-It should be clarified that only astrocytes *gfap*⁺ are being evaluated when analyzing how the loss of *Alk5* affects microglial or astrocytic morphology/gene signature.*

- We have specified that for astrocytic *Alk5* knockout, we used the *mGfapCre* driver. This mouse line has been reported to target more than 70% of total astrocytes starting from the neonatal stage to adulthood (PMID 15494728 and PMID 31194676).

- The sex of the animals used in the experiments should be mentioned. Microglial biology is extremely influenced by sex and this must be taken into account to discuss the results.

- The sex of each animal is now indicated on each main figure (males are triangles, females are circles), and we now indicate in the figure legends that both males and females were used for

all experiments. We observed no obvious differences between males and females in the analyses reported in our study.

- Please add the age of the animals that were used.

- As suggested, the age of all animals is now indicated in all figures.

- Information about the brain areas that were analyzed is needed. Particularly in the imaging section, please add information of which part of the cortex and hippocampus were used, specify the Bregma that was analyzed and clarify from which particular area the representative pictures shown were taken. This would be helpful to understand if the brain area influences the results.

- The phenotypes are global in the CNS. We presented the somatosensory cortex and hippocampus as representative regions. Information on the brain area analyzed has been added to each figure and the bregma coordinates are provided in the methods section.

Reviewer #4 (Remarks to the Author):

REVIEWERS' COMMENTS

Reviewer #1 (Remarks to the Author):

The authors have addressed all of the issues being raised by the reviewer. The manuscript has been improved significantly and will contribute essential pieces of information to the scientific community working with microglia. I have no further comments or suggestions.

Reviewer #2 (Remarks to the Author):

The authors have been quite responsive to the Reviewer critiques. Additional data have been added and modifications have been made to the text. The study is quite comprehensive in the analysis of Tgfb1 control of microglial functions in the brain.

Reviewer #3 (Remarks to the Author):

The authors have addressed all the comments and suggestions proposed by the reviewers. The manuscript was greatly improved.

Reviewer #4 (Remarks to the Author):

Response to Reviewers' Comments

We would like to thank the reviewers for their valuable comments and suggestions that have helped us substantially improve our manuscript further. Below is our point-by-point response to the reviewers' comments.

Reviewer #1 (Remarks to the Author):

The authors have addressed all of the issues being raised by the reviewer. The manuscript has been improved significantly and will contribute essential pieces of information to the scientific community working with microglia. I have no further comments or suggestions.

No response necessary.

Reviewer #2 (Remarks to the Author):

*The authors have been quite responsive to the Reviewer critiques. Additional data have been added and modifications have been made to the text. The study is quite comprehensive in the analysis of *Tgfb1* control of microglial functions in the brain.*

No response necessary.

Reviewer #3 (Remarks to the Author):

The authors have addressed all the comments and suggestions proposed by the reviewers. The manuscript was greatly improved.

No response necessary.

Reviewer #4 (Remarks to the Author):

No response necessary.